# Multiple plant diversity components drive consumer communities across ecosystems

Andreas Schuldt [1,2,3], Anne Ebeling[4], Matthias Kunz [5], Michael Staab [6,7], Claudia Guimarães-Steinicke[8], Dörte Bachmann[9], Nina Buchmann [9], Walter Durka [1,10], Andreas Fichtner [11], Felix Fornoff [6], Werner Härdtle[11], Lionel R. Hertzog[12], Alexandra-Maria Klein[6], Christiane Roscher [1,13], Jörg Schaller [14], Goddert von Oheimb [1,5], Alexandra Weigelt[1,8], Wolfgang Weisser[15], Christian Wirth[1,8], Jiayong Zhang [16], Helge Bruelheide [1,2] & Nico Eisenhauer [1,17]

Humans modify ecosystems and biodiversity worldwide, with negative consequences for ecosystem functioning. Promoting plant diversity is increasingly suggested as a mitigation strategy. However, our mechanistic understanding of how plant diversity affects the diversity of heterotrophic consumer communities remains limited. Here, we disentangle the relative importance of key components of plant diversity as drivers of herbivore, predator, and parasitoid species richness in experimental forests and grasslands. We find that plant species richness effects on consumer species richness are consistently positive and mediated by elevated structural and functional diversity of the plant communities. The importance of these diversity components differs across trophic levels and ecosystems, cautioning against ignoring the fundamental ecological complexity of biodiversity effects. Importantly, plant diversity effects on higher trophic-level species richness are in many cases mediated by modifications of consumer abundances. In light of recently reported drastic declines in insect abundances, our study identifies important pathways connecting plant diversity and consumer diversity across ecosystems.

[1] German Centre for Integrative Biodiversity Research (iDiv) Halle-Jena-Leipzig, Deutscher Platz 5e, 04103 Leipzig, Germany. [2] Martin Luther University Halle-Wittenberg, Institute of Biology/Geobotany and Botanical Garden, Am Kirchtor 1, 06108 Halle, Germany. [3] Faculty of Forest Sciences, Forest Nature Conservation, Georg-August-University Göttingen, Büsgenweg 3, 37077 Göttingen, Germany. [4] Friedrich-Schiller-University Jena, Institute of Ecology and Evolution, Dornburger Strasse 159, 07743 Jena, Germany. [5] Institute of General Ecology and Environmental Protection, Technische Universität Dresden, PF 1117, 01735 Tharandt, Germany. [6] University of Freiburg, Nature Conservation and Landscape Ecology, Tennenbacher Straße 4, 79106 Freiburg, Germany. [7] University of Freiburg, Freiburg Institute of Advanced Studies (FRIAS), Albertstraße 19, 79104 Freiburg, Germany. [8] Systematic Botany and Functional Biodiversity, Leipzig University, Johannisallee 21–23, 04103 Leipzig, Germany. [9] ETH Zurich, Institute of Agricultural Sciences, Universitätsstrasse 2, 8092 Zurich, Switzerland. [10] Helmholtz Centre for Environmental Research - UFZ, Department of Community Ecology, Theodor-Lieser-Str. 4, 06120 Halle, Germany. [11] Leuphana University of Lüneburg, Institute of Ecology, Universitätsallee 1, 21335 Lüneburg, Germany. [12] Terrestrial Ecology Unit, Gent University, K.L. Ledeganckstraat 35, 9000 Gent, Belgium. [13] UFZ, Helmholtz Centre for Environmental Research, Physiological Diversity, Permoserstrasse 15, 04318 Leipzig, Germany. [14] University Bayreuth, Environmental Geochemistry, Bayreuth Center for Ecology and Environmental Research (BayCEER), Universitätsstraße 30, 95447 Bayreuth, Germany. [15] Technical University of Munich, Department of Ecology and Ecosystem Management, Terrestrial Ecology Research Group, Hans-Carl-von-Carlowitz-Platz 2, 85354 Freising, Germany. [16] Key Lab of Wildlife Biotechnology, Conservation and Utilization of Zhejiang Province, Zhejiang Normal University, 321004 Jinhua, Zhejiang Province, China. [17] Leipzig University, Institute of Biology, Deutscher Platz 5e, 04103 Leipzig, Germany. Correspondence and requests for materials should be addressed to A.S. (email: andreas.schuldt@forst.uni-goettingen.de)

Safeguarding biodiversity has become a key societal concern in the light of global environmental change[1–3] and declining numbers of insects and other organisms[4–6], particularly because biodiversity plays an important role in the provisioning of ecosystem services[7,8]. The nowadays common management of many ecosystems for only a few, selected species of primary producers contributes to changes in overall biodiversity that might prove detrimental to human well-being[6,9,10]. Managing for a higher diversity of plants has therefore been suggested as a way to mitigate such potentially negative consequences[11,12] and is expected to promote both biodiversity and ecosystem functioning at higher trophic levels[11,13]. This is because plant diversity provides essential resources and habitat for higher trophic-level organisms[14]. Yet, previous studies in forests, grasslands, and other ecosystems varied in their support for the assumption that plant diversity promotes the diversity of herbivores, predators, or parasitoids[13,15–18].

A thorough understanding of the extent to which biodiversity effects across trophic levels can be generalized is hampered by the fact that the underlying mechanisms are often not well resolved[14]. Previous studies have focused primarily on plant species richness as one component of plant diversity[13,15–17,19], although changes in the functional composition and functional diversity of plant communities may ultimately drive the effects of plant species richness on higher trophic levels[20]. Nevertheless, functional redundancy among plant species can lead to non-linear or a lack of relationships between plant diversity and higher trophic levels[21,22]. In addition, the physical structure of plant communities has been shown to influence the spatial distribution and complexity of habitats, microclimates, and species interactions[23–25]. However, we still have a limited understanding of how the structural diversity of plant communities (both in terms of vertical distribution and horizontal variation across space) contributes to overall plant diversity effects on higher trophic levels and how it potentially interacts with the effects of plant functional diversity.

A better mechanistic understanding of how plant diversity affects the diversity and functioning of higher trophic levels may, therefore, be achieved by simultaneously considering and disentangling the relative contribution of plant functional and plant structural diversity to overall biodiversity effects[26,27]. Such an approach might also help to explain the variability in biodiversity effects among different ecosystems, such as grasslands and forests, if we were able to reduce complex plant diversity effects to basic principles related to plant structure and functional diversity. For example, differences in the vertical or horizontal distribution of structural diversity of grassland plants and trees affect gradients of light availability and temperature[28] that can strongly influence arthropod communities[23–25]. In this context, it may be particularly important to consider plant diversity effects on the abundances of higher trophic-level organisms, such as arthropods. The biomass and abundance of arthropods have recently been reported to decline significantly due to anthropogenic activities[4,6,29]. At the local scale, part of this decline may be due to changes in the structural and functional composition of plant communities[9,29]. However, the linkages between changes in plant diversity, changes in arthropod abundances, and the consequences for arthropod diversity (i.e., indirect effects of plant diversity that modify arthropod diversity via changes in arthropod abundances) at the scale of local communities are not yet well understood[5].

Here we disentangle the impact of changes in major components of plant diversity on species richness of herbivores, predators, and parasitoids for two different ecosystems. We make use of an extensive data set with 53 plant species and 34,060 individuals of 882 arthropod species of two large-scale biodiversity experiments, one in temperate grasslands[30,31] and one in subtropical forests[32]. This comparison can help us to obtain first insights into the extent to which effects of plant diversity might operate in similar ways in contrasting ecosystems. We use path models to analyze the relative contribution of direct and indirect effects of plant taxonomic diversity (species richness), functional diversity and composition, and structural diversity on overall arthropod species richness and the species richness of major trophic groups of arthropods. We quantified functional diversity as the variability among plant species in morphological and chemical leaf traits that were shown previously to affect arthropods[22,33,34]. Because plant traits can further influence arthropods via mass-ratio effects[35], we also tested for the effects of mean trait values on arthropod abundance and species richness. Vertical stratification and horizontal variation of plant height within study plots were used to quantify plant structural diversity. Importantly, we explicitly differentiated between direct and indirect plant diversity effects on arthropod species richness. We considered direct effects as those directly linking plant diversity to arthropod species richness (e.g., because plant diversity-mediated habitat diversity provides more niches that support a higher diversity of arthropods[36]). Because we hypothesized that arthropod species richness is influenced by changes in arthropod abundances (i.e., assuming that species richness is affected via more individuals[37]), we considered effects of plant diversity that modified arthropod abundances as indirect effects on arthropod species richness. Our study therefore provides important insights into the potential mechanisms linking changes in plant communities to consumer diversity via changes in abundances. We also tested the alternative hypothesis of reciprocal interactions between arthropod species richness and abundance[38], which might be better reflected by residual covariance terms than by a directional pathway in the path models. We show that the combination of plant functional and structural diversity mechanistically explains plant species richness effects on higher trophic levels in both ecosystem types. Although the relative effects of functional and structural diversity on arthropods differed among trophic levels and ecosystems, they operated in many cases via modifying arthropod abundances—indicating a high vulnerability of arthropod diversity to currently observed declines in arthropod numbers.

## Results

**Species composition across trophic levels.** In total, we sampled 8075 arthropods belonging to 506 (morpho)species (excluding singletons, i.e., species that only occurred with one individual) in the BEF-China forest biodiversity experiment. Herbivores were the most abundant and species-rich of the trophic groups we considered in our analyses (2204 individuals [27% of total arthropod abundance], 233 species [46% of total arthropod species richness]), followed by predators (1739 individuals [22%], 171 species [34%]) and parasitoids (617 individuals [8%], 32 species [6%]). In the Jena Experiment in grassland, we sampled 25,985 arthropods belonging to 376 species (excluding singletons). Predators were the most abundant (15,702 [60% of total]) and species-rich (184 species [49% of total]) group, followed by herbivores (6099 individuals [23%], 129 species [34%]) and parasitoids (1171 individuals [5%], 26 species [7%]). In both experiments, Pearson correlations between abundance and/or richness values of herbivores, predators, and parasitoids were always positive when significant ($P \leq 0.05$; Supplementary Table 1).

**Functional and structural diversity explain richness effects.** In both ecosystem types, plant species richness promoted leaf trait

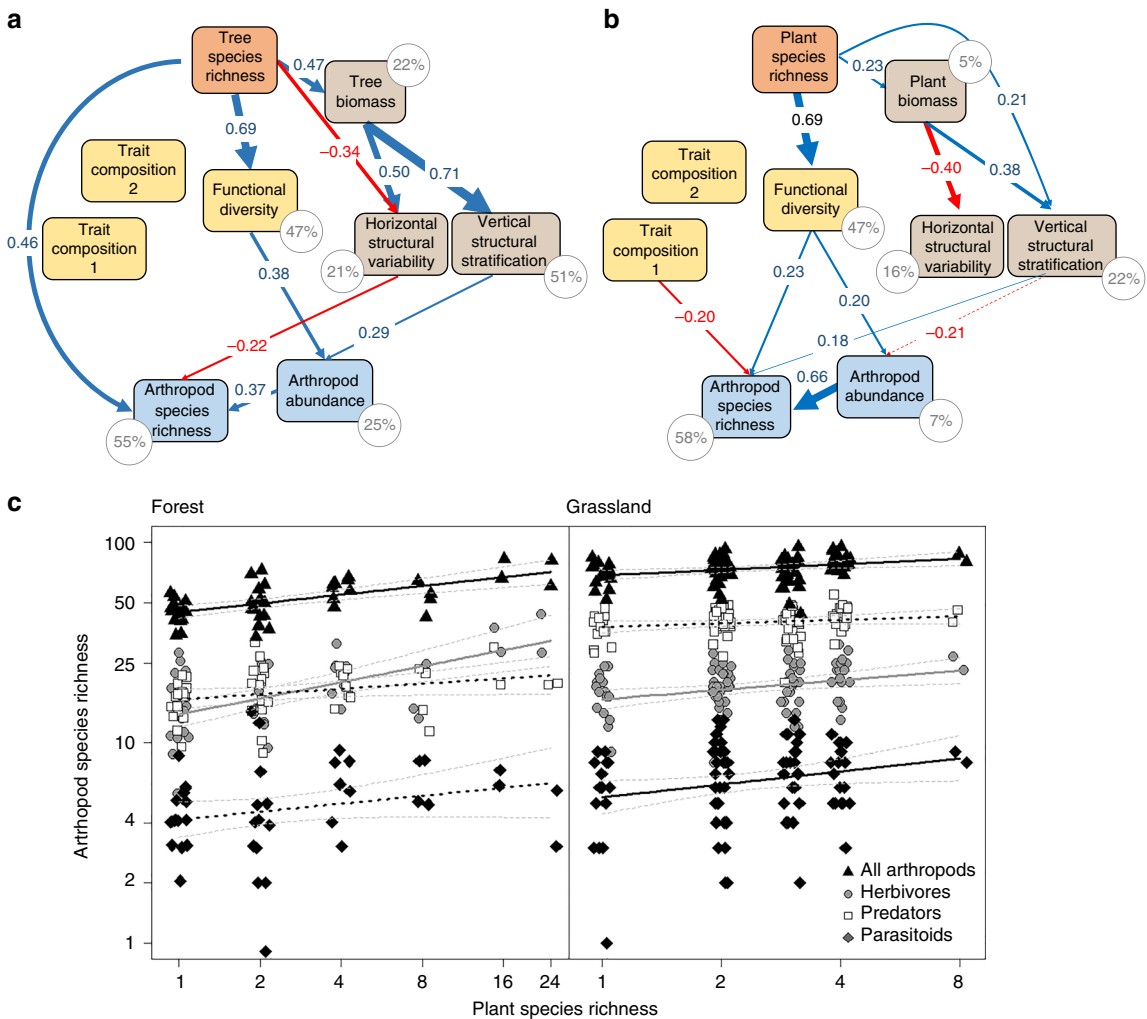

**Fig. 1** Effects of plant diversity on arthropod species richness. Direct and indirect effects of plant species richness (orange), leaf trait functional diversity and composition (yellow), and structural characteristics (brown) of the plant communities on overall arthropod abundance and species richness (blue) for **a** the forest system ($\chi^2 = 6.96$, DF = 10, $P = 0.789$), and **b** the grassland system ($\chi^2 = 19.9$, DF = 14, $P = 0.185$) based on path model results. Trait compositions 1 and 2 represent the first two axes of a principal components analysis (PCA) on community-weighted means of five leaf traits; functional diversity is the mean pairwise dissimilarity (based on Rao's $Q$) of these traits among study plots. Vertical stratification (based on Rao's $Q$) and horizontal variability of plant structure (based on Moran's $I$) represent variability in plant height (grassland) or the first two axes of a PCA on the variability of tree height and crown projection area (forest) within the study plots. Positive and negative pathways and their corresponding standardized path coefficients in **a** and **b** are indicated in blue and red, respectively. Solid lines show significant relationships ($P \leq 0.05$ based on 1000 bootstrap draws; scaled by their standardized effect size), dotted lines show non-significant pathways (see Supplementary Tables 2, 3 for full results). For clarity, covariances between structural and functional diversity (see Fig. S4) were not plotted but are shown in Supplementary Tables 2 and 3. Percentage values show the explained variance of endogenous (dependent) variables. Effects of plant species richness on arthropod species richness **c** in the BEF-China forest experiment (left panel; values corrected for the number of trees sampled; $N = 46$ study plots), and in the Jena Experiment in grassland (right panel; $N = 92$). Solid regression lines indicate significant ($P < 0.05$) relationships, broken regression lines show marginally significant relationships ($P < 0.1$). Broken gray lines are 95% confidence intervals. Source data are provided as a Source Data file

functional diversity (calculated as mean pairwise trait dissimilarity; Fig. 1a, b). Plant species richness also increased plant biomass and influenced—either directly, or indirectly via effects on biomass—plant structural diversity (calculated as pairwise dissimilarity and spatial dispersion in height; Fig. 1a, b). Plant trait composition (based on community-weighted mean (CWM) trait values), in turn, was not significantly affected by plant species richness in either ecosystem (Fig. 1a, b).

In both ecosystem types, plant species richness showed a significantly positive relationship with overall arthropod species richness (forest: $0.14 \pm 0.03$ SE, $F_{1,44} = 26.2$, $P < 0.001$; grassland: $0.09 \pm 0.03$ SE, $F_{1,90} = 8.4$, $P = 0.005$ for a linear regression on log-transformed species richness data) and with the species

richness of herbivores (forest: $0.27 \pm 0.06$ SE, $F_{1,44} = 20.2$, $P < 0.001$; grassland: $0.15 \pm 0.06$ SE, $F_{1,90} = 7.2$, $P = 0.009$ for a linear regression on log-transformed species richness data) (Fig. 1c). Likewise, predator and parasitoid species richness showed a marginally positive relationship with plant species richness in both the forest ($0.09 \pm 0.04$ SE, $F_{1,44} = 3.9$, $P = 0.055$ and $0.14 \pm 0.08$ SE, $F_{1,44} = 3.0$, $P = 0.090$, for linear regressions on predator and parasitoid richness, respectively) and the grassland system ($0.06 \pm 0.03$ SE, $F_{1,90} = 3.2$, $P = 0.076$ and $0.22 \pm 0.10$ SE, $F_{1,90} = 4.8$, $P = 0.030$, respectively) (Fig. 1c). The associations between plant species richness and arthropod species richness were to a large extent explained by plant functional and structural diversity in both ecosystems (Figs. 1a, b, 2, and 3). Only forest

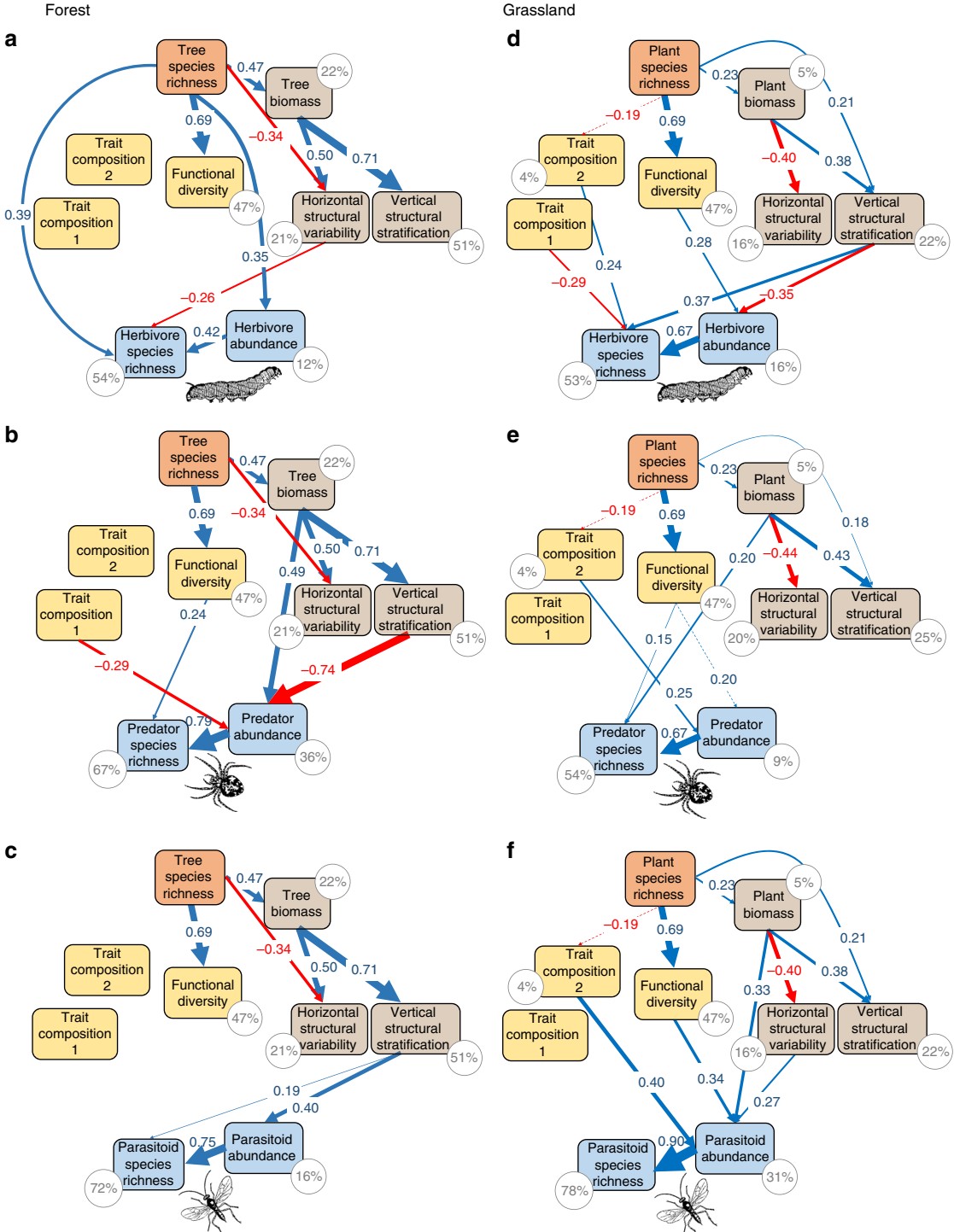

**Fig. 2** Effects of plant diversity on arthropods across trophic levels. Direct and indirect effects of plant species richness (orange), leaf trait functional diversity and composition (yellow), and structural characteristics (brown) of the forest and grassland plant communities on arthropod abundance and species richness (blue) based on path model results. **a** Forest herbivores ($\chi^2 = 6.6$, DF = 11, $P = 0.796$), **b** forest predators ($\chi^2 = 13.4$, DF = 17, $P = 0.723$), **c** forest parasitoids ($\chi^2 = 7.7$, DF = 12, $P = 0.814$), **d** grassland herbivores ($\chi^2 = 25.3$, DF = 20, $P = 0.259$), **e** grassland predators ($\chi^2 = 14.8$, DF = 15, $P = 0.495$), **f** grassland parasitoids ($\chi^2 = 27.2$, DF = 21, $P = 0.302$). Trait compositions 1 and 2 represent the first two axes of a principal components analysis (PCA) on community-weighted means of five leaf traits; functional diversity is the mean pairwise dissimilarity (based on Rao's $Q$) of these traits among study plots. Vertical stratification (based on Rao's $Q$) and horizontal variability of plant structure (based on Moran's $I$) represent variability in plant height (grassland) or the first two axes of a PCA on the variability of tree height and crown projection area (forest) within the study plots. Positive and negative pathways and their corresponding standardized path coefficients in **c** and **d** are indicated in blue and red, respectively. Solid lines show significant relationships ($P \leq 0.05$ based on 1000 bootstrap draws; scaled by their standardized effects), dotted lines show non-significant pathways (see Supplementary Tables 4–9 for full results). For clarity, covariances between structural and functional diversity (see Fig. S4) were not plotted but are shown in Supplementary Tables 4–9. Percentage values show the explained variance of endogenous (dependent) variables. The animal icons (from www.openclipart.org) are licensed for use in the public domain without copyright (Creative Commons Zero 1.0). Source data are provided as a Source Data file

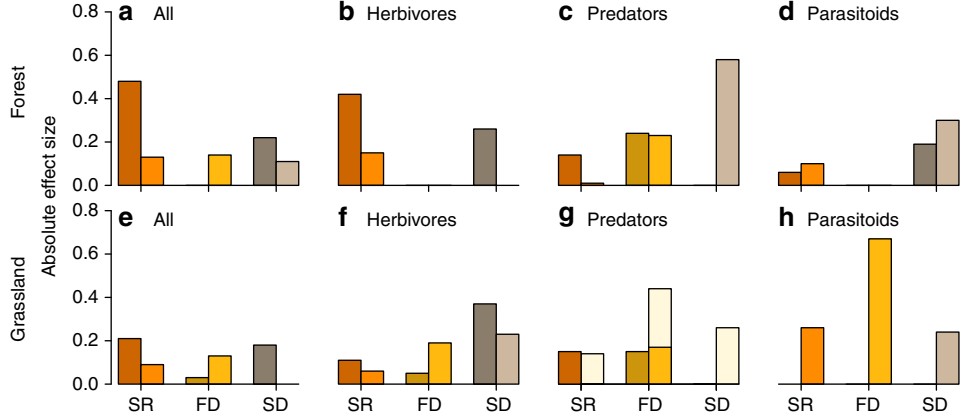

**Fig. 3** Direct and indirect effects of plant diversity on arthropod species richness. Bars show summed effects of plant taxonomic (SR, orange color), functional (FD, yellow color), and structural (SD, brown color) diversity on arthropod species richness, obtained from the path models in Figs. 1 and 2. Absolute values are shown to enable better comparison of effect sizes among predictors. Effects are either direct (darker hues, left bar of each diversity component, connecting arthropod richness with plant diversity via direct paths) or indirect via arthropod abundance (lighter hues, right bar of each diversity component, effects of plant diversity on arthropod abundance, which in turn affected arthropod species richness). **a–d** show results for the forest experiment, **e–h** for the grassland experiment, with **a**, **e** featuring overall arthropod species richness, **b**, **f** herbivores, **c**, **g** predators, and **d**, **h** parasitoids. beige-colored bars in **g** show indirect effects of plant diversity on grassland predator species richness when restricting the analyses to arthropods directly sampled from the vegetation (i.e., excluding data from pitfall traps). Absolute effect size was calculated as the product of standardized path coefficients connecting each plant diversity component with arthropod species richness, summed over the individual predictors of each component (i.e., trait composition 1+trait composition 2+functional diversity for FD and horizontal structural variability+vertical structural stratification for SD). Note that direct and indirect effects of plant species richness (SR) on arthropod richness also include the effects via FD and SD (because SR influenced FD and SD, and effects of FD and SD on arthropod richness are therefore partial effects of SR). Source data are provided as a Source Data file

herbivores showed a direct relationship with plant species richness, a pattern that was also found for overall arthropod species richness because of the large proportion of herbivores (Figs. 1a and 2a, indicating mechanisms not addressed by our study design and selection of predictor variables).

In both study systems and for all trophic levels (herbivores, predators, parasitoids), arthropod abundances had strong positive effects on arthropod species richness (standardized path coefficients ranging between 0.42 and 0.90), and significant associations between plant diversity and arthropod species richness were often indirect via effects on arthropod abundances (Figs. 1, 2, and 3). However, the relative influence of plant functional and structural diversity on arthropod abundances and arthropod species richness differed between the two study systems (Fig. 2). Results for overall arthropod diversity in the forest ecosystem strongly reflected the impact of plant diversity on the abundance and species richness of the dominating herbivores (Figs. 1a and 2a). In contrast, results for overall arthropod diversity in the grassland ecosystem reflected a mix of the relationships between plant diversity and both predator and herbivore abundance and species richness (Figs. 1b and 2d, e). In general, directional effects of arthropod abundance on arthropod species richness were more strongly supported than interdependent effects (expressed as residual covariance terms) between abundance and species richness. In both ecosystems, second-order Akaike Information Criterion (AICc) values were lowest for models assuming that arthropod abundance drives patterns in arthropod species richness for overall, predatory, and parasitoid arthropods (Supplementary Tables 2–17). Only in the case of forest and grassland herbivores were models assuming more complex interdependencies between arthropod abundance and species richness equally likely (Supplementary Tables 11 and 15). Models based on rarefied arthropod species richness showed that when factoring out arthropod abundance, many of the indirect and direct relationships between plant diversity and arthropod species richness disappeared (Supplementary Tables 18–24),

underpinning the role of arthropod abundance changes in modifying the relationships between plant and arthropod diversity.

**Strength of diversity effects varies across trophic levels.** In the forest system, herbivore species richness was not significantly related to the functional identity of the tree communities (as described by weighted trait means) and was only moderately related to plant structural diversity (negative effect of horizontal variation of tree structure; Fig. 2a). Instead, forest herbivore species richness and abundances showed a strong, positive relationship with tree species richness. Associations between structural diversity and arthropod abundance or species richness in the forest system became stronger at higher trophic levels (standardized path coefficients for herbivores −0.26 for horizontal structural variation, compared to −0.74 for vertical stratification for predators, and 0.19 (horizontal) and 0.40 (vertical) for parasitoids). The influence of structural diversity dominated the overall (direct and indirect) relationships between tree diversity and the abundance and species richness of predators and parasitoids (Fig. 3). The vertical stratification of tree height and crown size (based on crown projection area) was strongly negatively related to predator abundances and positively to parasitoid abundances (Fig. 2b, c). Moreover, forest parasitoid species richness increased with the horizontal variation of tree structure (Fig. 2c). Predator abundance strongly increased with tree biomass (Fig. 2b). Relationships between tree functional diversity or composition and arthropods in the forest system were weaker and most pronounced for predators: their species richness increased and their abundance decreased with increasing functional diversity and mean trait values (PC1 related to high leaf toughness and low specific leaf area (SLA) and leaf nitrogen concentration; Supplementary Table 25) (Fig. 2b).

In the grassland system, plant functional diversity was consistently positively related to arthropod abundances across

trophic levels (Fig. 2d–f). Trait composition representing the dominance of plant species with high leaf toughness and silica concentrations and low leaf nitrogen concentrations (PC1; Supplementary Table 26) were associated with decreased grassland herbivore species richness. Trait composition related to the dominance of plant species with low SLA and leaf carbon concentrations (PC2; Supplementary Table 26) showed a positive relationship with herbivore species richness and both predator and parasitoid abundance in the grassland system (Fig. 2d–f). Plant biomass was directly associated with predator species richness and parasitoid abundance (Fig. 2e, f). Vertical stratification and horizontal variation in plant structure particularly influenced herbivores (Fig. 2d) and, to a lesser extent, parasitoids (Fig. 2f). Predators in grassland were only significantly related to plant structure when excluding the majority of ground-active species (i.e., focusing on suction samples: positive effect of vertical stratification of plant structure on predator abundance; Fig. 3g, Supplementary Fig. 1).

## Discussion

Our study shows for two contrasting ecosystems that functional and structural diversity of the plant communities strongly contribute to explaining the positive relationships between plant species richness and the plot-level species richness of invertebrate consumers across trophic levels. The relative importance of plant functional and plant structural diversity differed across trophic levels and between ecosystems. Yet, many strong associations between plant diversity and arthropod species richness were consistently found to operate via relationships with arthropod abundances. These findings have important implications for attempts to develop a more detailed understanding of biodiversity relationships and the impact of global environmental change across trophic levels, and they highlight important avenues for the future of biodiversity research.

First, our results indicate that structural diversity metrics of plant communities are highly relevant mediators of plant diversity effects on arthropod diversity and that they strongly contribute to a mechanistic explanation of these effects. Recently, researchers have started to address the mechanisms underlying previously observed effects of plant species richness on higher trophic levels by testing for the potential role of plant functional diversity and composition[22,39,40]. However, structural diversity as an additional mediator of plant diversity effects on consumer diversity has received much less attention. This is despite the well-known fact that plant structure significantly affects herbivores, predators, and parasitoids by modifying environmental conditions and habitat space[23–25] and that plant species richness can influence the physical structure of plant communities[41,42].

Associations with plant structural diversity were particularly pronounced for forest arthropods. This might be explained by the size and longevity of trees compared to grassland plants. Trees function as keystone structures that ensure long-term habitat continuity for associated arthropods, while mowing of grasslands (two times per year in our grassland study system) leads to seasonal changes in vegetation structure (with consequences for arthropod community composition, as reflected by differences in the frequency and identity of dominant arthropod taxa; see Methods). Moreover, the large size of the trees compared to grassland plants results in spatially more extensive microclimatic gradients from light-exposed upper canopy parts to shaded interiors of the canopy[28]. These differences might explain why the resulting relationships with plot-level arthropod species richness were more important than differences in leaf functional characteristics in the forest system.

The consistent associations with leaf trait functional diversity at all trophic levels (herbivores, predators, parasitoids) in the grassland system might be indicative of bottom–up effects that propagate through the food web from plants via herbivores to predators and parasitoids, consistent with previous results reported for the effects of plant species richness in both study systems[16,22]. Differences in sampling methods between the study systems probably play a minor role: forest arthropods were all sampled directly from the vegetation (by beating), as were most grassland herbivores and parasitoids (primarily captured by suction sampling). Moreover, although most grassland predators were ground-active (sampled with pitfall traps), restricting the analyses to predators sampled from the vegetation (for which plant structure might be more important than for ground-active arthropods) did not change the relative importance of plant functional vs. structural diversity effects.

At the same time, however, the relative importance of functional and structural diversity on arthropods varied substantially across trophic levels in both ecosystems. While studies replicated across a wider range of environmental conditions and manipulative experiments will be required to verify the causal drivers and generality behind the observed effects, this variability across trophic levels provides indications of potential mechanisms. Negative relationships between vertical stratification or horizontal variation in plant structure and herbivore abundances could indicate a disruption of host-finding abilities or of herbivore dispersal in structurally more heterogeneous environments[27,43]. In contrast, direct positive associations with herbivore and parasitoid species richness might reflect a higher diversity of habitats and niches for different species[23,24].

While the lack of relationships between leaf trait functional diversity and forest herbivores could indicate that functional traits not considered in our study play a role, strong correlations between functional diversity and phylogenetic diversity (often used as a metric to capture unmeasured variability in functional traits[44], see Methods) suggest that the traits used in our analyses account for an important part of the overall trait space. In our case, the results might therefore suggest that tree functional trait effects on higher trophic levels did not primarily act via trophic linkages. This implies that tree diversity operated via direct effects on predators and parasitoids, and not via the modification of prey abundance and species richness[45]. This is in line with the assumption of the enemies hypothesis that effects of plant diversity on predator diversity can also operate via modifications of habitat structure or reduction of intra-guild predation[14,23].

In the case of plant functional characteristics, such direct effects might be related to fine-scale structures—expressed at the level of leaves—that correlate with functional traits[46]. Many of the forest predators were web-building spiders[22]. Differences in leaf toughness or SLA (as represented by principal components analysis (PCA) axis 1 of trait composition, which negatively affected forest predator abundance) might affect leaf structural attributes that are important for the diversity of possible web-attachment points and which therefore influence predator species richness[23]. Moreover, the abundances of these predators might be more strongly promoted by the total availability of habitat space, as indicated by the strong positive effects of tree biomass on forest predator abundance. Variability in tree size might reduce the overall availability of habitat space for dominant species with specific habitat requirements, which could explain the negative effect of vertical stratification of tree structure on forest predator abundance. Nevertheless, our finding that plant biomass effects on arthropods often worked indirectly via structural diversity shows that structural diversity can also be important for mechanistically understanding the consequences of diversity–productivity relationships for

ecosystem functioning[47]. In the case of grassland herbivore species richness, the negative effect of leaf silica concentrations (represented by PCA axis 1 of trait composition) might be indicative of the previously suggested role of silica as a defense against herbivores[48].

Second, our finding that relationships between plant diversity and arthropod species richness were in many cases indirect via the modification of arthropod abundances has important consequences for our ability to predict biodiversity change in response to global environmental change. Strong and consistent effects of arthropod abundance on arthropod species richness suggest an important role of pathways related to the more-individuals hypothesis (i.e., more individuals allow for viable populations of more species[37,38]). The interesting finding is that of the many possible pathways and mechanisms that potentially link plant diversity to higher trophic level diversity (many of which are direct effects between plant and animal diversity, e.g., via modifications of habitat diversity that supports a higher diversity of animals[39,49]), those that influence species richness via changes in abundance made an important contribution to explaining overall diversity effects in both study systems. These effects were in many cases as strong as or even stronger than the direct effects of taxonomic, functional, or structural plant diversity on arthropod species richness. The important mediating role of arthropod abundances on the relationships between plant diversity and arthropod species richness also became evident when factoring out arthropod abundances by rarefaction and when comparing models with direct pathways vs. covariation between arthropod abundance and species richness.

Recently, scientists and society have become increasingly aware of drastic declines in arthropod biomass[5,6,29]. However, the linkages between these declines and ongoing changes in biodiversity remain less clear[5,29]. In this context, our study helps to disentangle important pathways connecting changes in the environment and in biodiversity via species abundances. Our results underscore the importance of more thoroughly considering these linkages. Explicitly incorporating changes in species abundance and how these changes are mediated by environmental change can be critical to understanding current and future changes of biodiversity and associated ecosystem functions. In our study, these effects would have stayed elusive without the inclusion of plant structural diversity, highlighting the benefits of simultaneously considering multiple components of plant diversity and the potential mechanisms discussed above. The same may be true for higher trophic-level diversity and the diversity of interactions among trophic levels, and we hope that our study stimulates future research exploring such interactions. In particular, the top–down effects of predators and parasitoids on herbivores[14], cascading effects of plant diversity via herbivores on secondary consumers[16], or effects of other functional groups (e.g., insectivorous birds[50]) are additional modifiers that deserve further research and that our models take into account only implicitly by analyzing the net effect of plant diversity on individual trophic levels. Our findings are particularly important in the light of ongoing habitat simplification and the loss of structural heterogeneity of ecosystems[51], and they support management recommendations that aim at maintaining and increasing the structural diversity of ecosystems (e.g., promoting uneven-aged forests[52] and grazing regimes in grasslands[53]). At the same time, the variability in effects of plant functional and plant structural diversity on arthropod abundance and species richness across trophic levels and ecosystem types cautions against overly simplistic generalizations and underscores the necessity of future research to take the ecological complexity of ecosystems into account.

## Methods

**Study sites and experimental design.** We considered two large-scale and long-term plant diversity experiments representing a forest[32] and a grassland ecosystem[30], respectively.

The BEF-China forest experiment is located close to Xingangshan, Jianxi Province, China (29°08′–29°11′ N, 117°90′–117°93′ E, 100–300 m above sea level) and represents subtropical mixed evergreen broadleaved forest. The mean annual temperature at the study site is 16.7 °C, and mean annual precipitation is 1800 mm (ref. [32]). The experiment consists of two study sites (A and B) established in 2009 and 2010, respectively. It comprises 566 study plots of 25.8 × 25.8 m². Planted species richness, based on a pool of 40 broadleaved tree species, ranges from monocultures to mixtures of 2, 4, 8, 16, and 24 species. Trees were planted in a regular grid (20 rows and 20 columns) with 1.29 m planting distance among trees for a total of 400 trees per plot. Species were randomly assigned to individual planting positions within the plots, with the total number of individuals per plot divided equally among the species planted in a given plot[32].

Our analyses followed the design for a set of 64 (32 per site, randomly distributed across the sites) very intensively studied plots. Tree species composition of the mixtures was determined by randomly assigning (without replacement) each species of the 16-species mixtures to one 8-species mixture, subdividing these sets of 8 tree species to non-overlapping subsets of four species, and the 4-species subsets to non-overlapping 2-species mixtures[32]. The 24-species mixtures were included as an additional high diversity treatment, which contained an additional eight species not present in the other plots of the study site. Tree species composition differed between the two sites, with two separate species pools of 16 broadleaved species in each site and an additional 8 species shared between sites in the 24-species mixtures. All plots were weeded twice a year, with all upcoming vegetation between the planted trees being removed. Lack of or limited tree establishment (8 plots) and lack of arthropod sampling (10 plots, see below) limited the final set of plots to 46 (16 monocultures, 14 2-species mixtures, 8 4-species mixtures, 4 8-species mixtures, 2 16-species mixtures, and 2 24-species mixtures).

The Trait-Based Experiment (TBE), one of the experimental grassland experiments running in the framework of the Jena Experiment, is located close to Jena, Thuringia, Germany (50°55′N, 11°35′E; 130 m above sea level) and represents mesophilic temperate grasslands. The mean annual temperature at the study site is 9.9 °C, and mean annual precipitation is 610 mm (ref. [31]). The TBE was established in 2010 on a former arable land and comprises 138 study plots of 3.5 × 3.5 m². Sown plant species richness ranges from monocultures to mixtures of 2, 3, 4, and 8 species. The 20 plant species (grasses and non-legume herbs) sown in the experiment were selected from a set of 60 grassland species representing the whole species pool of the Jena Experiment, based on their degree of complementarity in 6 functional traits related to resource acquisition strategies[30]. Plant mixtures were assembled to represent varying degrees of plant functional diversity (four levels from low to high functional diversity based on the six selected plant traits) within species richness levels[30]. Plots were arranged in a randomized block design and are mown twice (according to the common management of extensively used hay meadows in the region) and weeded three times a year (to maintain the sown species composition). For our analyses, we used the 92 plots of the TBE (2 species pools of 8 species each, with full replication of the diversity gradient for each species pool) for which terrestrial laser-scanning data to determine vegetation structure were available.

**Arthropod sampling and species richness.** Arthropods were sampled in both experiments in 2014, using quantitative methods best suited for a representative assessment of their diversity in each ecosystem: branch beating, standardized assessments of trophobioses (mutualistic interactions between ants and hemipterans), and trap nests in the forest system; pitfall traps and suction sampling in the grassland system.

In the forest system, herbivorous and predatory arthropods were sampled from the trees by beating: arthropods were knocked down onto a white cloth sheet (ø 100 cm) by hitting the trees several times with a padded stick[22]. Sampling was conducted during two seasons of peak arthropod abundance (May and September 2014), using the first four rows of trees for a total of 40 planting positions in each plot. Arthropods were sorted in the laboratory, identified to family or genus level, and assigned to species or morphospecies. DNA barcoding of the cytochrome oxidase 1 was conducted following standard protocols[54] to verify our classification in potentially ambiguous cases (e.g., polymorphism, sexual dimorphism; see also ref. [55]). Data on ants and aphids were obtained from assessments of trophobiotic ant–aphid interactions conducted twice, in June/July and September/October 2014. For these assessments, trees in the core area of each plot were sampled[56]. Three branches per tree were randomly selected, and a total of 20 young leaves were visually inspected for the occurrence and the number of sap-sucking Hemiptera and honeydew-collecting ants[56,57]. Voucher specimens were collected and identified to the lowest possible taxonomic level. Parasitoid arthropods and their hymenopteran host species were sampled with standardized trap nests (polyvinyl chloride (PVC) tubes of 22 cm length and 12.5 cm diameter filled with reed internodes[58]. Trap nests fixed to wooden posts (1.5 m high) were exposed at two locations on each plot from September to December 2014. Internodes with nests of Hymenoptera were replaced monthly. Nests were brought to the laboratory and reared at ambient conditions until specimens hatched. Species were then identified

to species or morphospecies. The following taxa were considered in the analysis of the forest plots: Araneae, Blattodea, Orthoptera, Mantodea, Psocoptera, Hemiptera, Coleoptera, Hymenoptera (including parasitoids), parasitic Diptera, and Lepidoptera. We assigned species to functional groups (herbivores, predators including omnivores, parasitoids, others) based on published literature[55] and expert classification (Supplementary Data 1). Pollinators were not considered as a functional group in our analyses because the sampling methods employed did not allow for a consistent assessment and reliable comparison between the experiments.

In the grassland system, epigeic arthropods were sampled with pitfall traps. Traps consisted of plastic cups with an opening diameter of 4.5 cm, were filled with a 3% formaldehyde solution, and covered by a small roof as a rain shelter[30]. In the center of each plot, one trap was installed and kept running from the end of April until the beginning of September 2014. The traps were emptied and refilled at 14-day intervals. Arthropods in the vegetation were sampled by means of suction sampling with a modified vacuum cleaner (Kärcher A2500, Winnenden, Germany). Suction sampling was conducted twice, in May and July 2014. Per plot and sampling date, two patches of $0.75 \times 0.75$ m$^2$ were sampled by putting a gauze cage over the vegetation (to prevent arthropods from escaping) and removing all arthropods from the cage with the suction sampler[19]. Adult individuals belonging to the following taxa were then identified, as far as possible, to species level: Isopoda, Myriapoda, Chilopoda, Araneae, Orthoptera, Hemiptera, Hymenoptera (including parasitoids), and Coleoptera (Supplementary Data 1). We note that the set of organism groups considered in the two experiments is not identical. This is due to differences in arthropod communities of the two different types of ecosystems considered (forest vs. grassland), which nevertheless reflect the general composition of the dominant groups of herbivores, predators, and parasitoids in each ecosystem.

Data on arthropod species richness were pooled per plot across sampling methods, i.e., adding together total species numbers for all sampling methods to obtain plot-level data on the overall species richness of all arthropods, and of herbivores, predators, and parasitoids, respectively. Species occurring with only one individual in the entire sample (singletons) were excluded, as were organism groups in the beating data that were not sampled representatively by beating (e.g., Hymenoptera and Diptera) and for which sampling with other methods, such as suction sampling and trap nests, was considered more adequate. We removed singletons to make the data set more robust, because species recorded with just one individual in the whole data set might be vagrants that are not really associated with the respective study systems or the specific plots they were recorded in. While singleton species accounted for 13–49% of the total number of species across all study plots (forest: 47%, 48%, 49%, and 20% of all, herbivorous, predatory, and parasitoid species, respectively; grassland: 31%, 30%, 28%, and 13% of all, herbivorous, predatory, and parasitoid species, respectively), singleton removal did not influence overall patterns among study plots of arthropod species richness and abundance, which were highly correlated in the data sets with and without singletons (Pearson correlation, $r > 0.97$, $P < 0.001$ in all cases and for all trophic levels). Because missing or dead trees of some species affected the number of trees present in the part of the plots sampled for arthropods in the forest system (independent of tree species richness[22]), we regressed arthropod species richness over the number of trees sampled by beating in each plot and used the residuals as a sample size-corrected metric of species richness. We used the number of individuals (and for trophobiotic ants and aphids the occurrence, i.e., the number of trophobioses per tree[56,57]) pooled over all sampling methods as measures of plot-level abundance.

**Plant biomass.** We estimated overall plant biomass as a predictor of arthropod species richness and species-specific biomass per plot for the calculation of trait-based functional plant diversity. For the forest system, we used estimates of wood volume as a proxy of leaf biomass per tree, calculated from data on basal area and tree height assessed in October 2014 (ref. [59]). Assessments were based on the central $6 \times 6$ trees per plot (out of the grid of $20 \times 20$ trees planted in each plot) in monocultures and 2-species mixtures and the central $12 \times 12$ trees in more diverse mixtures. Values were upscaled to represent the total plot biomass. In the grassland system, plant biomass was assessed twice in 2014, at peak standing biomass in May and August. In each plot, all vegetation was clipped 3 cm above ground in two randomly selected areas of $20 \times 50$ cm$^2$. Samples were sorted to species level and weighed after drying for 72 h at 70 °C. Data were averaged across the two replicates per plot at each sampling date and then pooled across dates for an overall value of biomass production across the growing season.

**Plant functional traits and functional diversity.** We used a range of plant functional traits that characterize the nutritional quality of leaves and that have generally been found to influence arthropod (in particular herbivore) abundances and species richness[34,39]. These traits comprised SLA, leaf dry matter content (LDMC), leaf nitrogen (N) concentration, leaf carbon (C) concentration, and leaf toughness. These traits have repeatedly been found to explain a large proportion of the variation in arthropod diversity, community structure, or functioning in the ecosystems studied here as well as in other ecosystems, for example, because they influence leaf palatability[22,33,34,39]. For the grassland experiment, we additionally considered leaf silicon concentration, because its presence particularly in grasses

can significantly affect herbivores[60]. For both experiments, we used mean trait values per plant species as the average of trait measurements on individual plants, because plot-level data were not available for any of the traits in the forest experiment and for several of the traits measured in the grassland experiment.

Trait measurements followed standard protocols[61]. In the forest plots, traits were measured on sun-exposed leaves of a minimum of five plant species (ref. [62]). In the grasslands, bulk samples composed of 5–10 fully expanded leaves from a least three different individuals were collected in each plot, where the species occurred in the sown species combinations, for measurement of SLA, LDMC, N, and C in May and August 2012. We averaged trait values per species across the two measurement campaigns in the grassland experiment. Data on leaf toughness was not directly available from plants grown in the field but measured for five healthy and fully developed leaves on each of five replicate individual mesocosm plants (see ref. [63]), grown in PVC pipes (15 cm diameter, 60 cm length) filled with sieved field soil from the Jena Experiment mixed with 20% sand. Leaf toughness was measured as leaf penetration persistence at the center of the leaf blade in a stripe of 1 cm distance to the central vein using an electric penetrometer (force gauge FH50, Sauter GmbH, Germany, equipped with a 1.4 mm diameter metal needle). Silicon concentration was determined from species-specific biomass samples taken from 2005 to 2007 on the main experimental plots of the Jena Experiment[64]. We used a microwave digestion system (CEM Corporation, Matthews, NC, USA) for measurements. Ground plant material was digested at 180 °C using 3 ml HNO$_3$, 2 ml H$_2$O$_2$, 0.5 ml HF, and 5 ml H$_3$BO$_3$. Afterwards, silicon was determined by inductively coupled plasma optical emission spectroscopy (Optima 7000DV, Perkin Elmer) with ultraviolet detection and quantification at 251.6 nm (Si)[64].

Functional leaf trait diversity was calculated from multiple traits as Rao's quadratic entropy $Q$[65], which we used to quantify the mean pairwise dissimilarity among the plant species growing in a study plot based on the above traits. Trait values were weighted by the biomass data of each plant species in each plot (see above). For each trait, we further calculated CWM values (ref. [66]) as the biomass-weighted average of each trait per plot. While effects of CWM indicate mass-ratio effects of functional trait means (functional composition), Rao's $Q$ quantifies the variation around this mean and therefore indicates effects of trait variability[35] (functional diversity). To reduce the dimensionality of the CWM data, we subjected the CWMs of the individual traits to a PCA. For both ecosystems, this yielded two principal components (PCs) that captured together 66 and 70%, respectively, of the overall variation in trait composition of the two experiments (Supplementary Tables 25 and 26). In both systems, increasing values of PC1 reflected increasing leaf toughness and decreasing leaf nitrogen concentrations, while PC2 reflected decreasing leaf carbon concentrations (Supplementary Tables 25 and 26).

Because analyzing selected traits might not necessarily capture the full variation in functional diversity, we additionally calculated plant phylogenetic diversity. Phylogenetic diversity might be used as a proxy of overall functional trait space if functional traits show a phylogenetic signal[44]. We used ultrametric phylogenetic trees available for both experiments[67,68] (Supplementary Fig. 2) and calculated phylogenetic diversity, analogous to functional diversity, as biomass-weighted Rao's $Q$. However, functional and phylogenetic diversity were highly correlated in both experiments (Pearson correlation, $r = 0.83$, DF = 44, $P < 0.001$ for the forest experiment, and $r = 0.86$, DF = 90, $P < 0.001$ for the grassland experiment, based on log-transformed values). The same applied to the relationship between phylogenetic diversity and plant species richness (Pearson correlation, $r = 0.82$, $P < 0.001$ in forest and $r = 0.76$, $P < 0.001$ in grassland, DF as above), whereas functional diversity was less strongly correlated with plant species richness (Pearson correlation, $r = 0.69$, $P < 0.001$ in both experiments, DF as above). To avoid overly complex models, we therefore did not include phylogenetic diversity in our analyses, as its variation was already well reflected by functional diversity and plant species richness.

**Structural diversity.** We derived two metrics of plant structural diversity at the plot level. The metrics indicate (i) the vertical stratification of plant height (and for trees: crown size) per plot and (ii) the horizontal variation of this stratification across each plot (Supplementary Fig. 3). These metrics were based on the spatial variability in plant height (and additionally for trees: crown size, quantified as crown projection area) as general indicators of structural diversity at the plot level. Plant height and crown size are related to the stratification of foliage[41]. Their spatial distribution (both vertically from the ground upward and horizontally in terms of spatial variation) influences important habitat features of arthropods, such as microclimate, availability of food, shelter, or habitat space (e.g., web-attachment points for web-building spiders[23,24]). Analyses on the spatial variability of these indicators within study plots may therefore provide information on the heterogeneity in the availability and spatial arrangement of habitats and resources.

Plant height and (for trees) crown projection area were measured in 2014. In the forest system, tree height and crown projection area were measured directly with measuring tapes in September and October 2014. Measurements were conducted on the central $6 \times 6$ (monocultures and 2-species mixtures) or $12 \times 12$ (more diverse mixtures) tree individuals in each plot. Tree height was quantified as the total length [cm] from stem base to apical meristem. Crown projection area was calculated as the area spanned by an ellipse connecting horizontal crown diameter

measurements in two cardinal directions[69]. In the grassland system, measurements were conducted with the terrestrial laser scanner (TLS) Faro Focus 3D X330 (FARO Technologies Inc.). We scanned 92 plots before harvest in May 2014 at the peak of standing biomass. The TLS was mounted on a tripod in an upside-down position elevated 3.35 m above soil level. The scanner operates with a wavelength of 1550 nm and captures fully three-dimensional information of the plants allowing to extract accurate height measurements and spatial distributions at the mm level. Scans were performed with a scan resolution of 0.25 (corresponding to spatial resolution 3 mm at 3.35 m distance; see FARO Focus manual). For each plot, we extracted an area of 3.75 m$^2$ ($1.5 \times 2.5$ m$^2$) below the scanner to reduce the effect of shadowing within scans. Individual scans of each plot were cleaned using standard stray filters and transformed from a point cloud into $XYZ$ coordinates by using the proprietary software Scene (version 5.2.0, Faro Technologies, Inc., Lake Mary, Florida, USA). In addition, the point clouds were cleaned using a statistical outlier removal filter ($N = 6$, Sigma $= 1.5$) in the CloudCompare software (version 2.6). Plant height and variation of height were computed at a 5-cm grid interval. This corresponds to $50 \times 30$ grid cells on average in the observed area and is sufficient for capturing small-scale structural variability of individual grassland plants.

Vertical stratification of plant structure was quantified as the mean pairwise dissimilarity in plant height (and for trees: crown projection area) among all individual trees (forest) or 5 cm grid cells (grassland) per study plot, calculated as Rao's $Q$[65]. Horizontal variation of plant structure was calculated as the spatial variation in plant height (and for trees: crown projection area) within each study plot based on Moran's $I$[70]. Values of Moran's $I$ close to 0 indicate a spatially random distribution of the variable of interest, while lower and higher values indicate spatial dispersion of dissimilar values and spatial clustering of similar values, respectively. We therefore interpreted increasing values of Moran's $I$ as a trend toward increasing spatial aggregation of structurally similar plants within the study plots, which we considered as indicative of lower horizontal structural diversity at the plot level. We used inverse distance weighting for the computation of Morans' $I$, assuming reduced spatial dependence with increasing distances between individual plants. Dead trees and gaps without plants were assigned a height of 0 cm.

Because crown projection area increased with tree height (Pearson correlation, $r = 0.73$; $P < 0.001$), we subjected the metrics of Rao's $Q$ and Moran's $I$ for the forest system to a PCA. This yielded two orthogonal principal components (explaining 81% of the total variation in the data), the first one reflecting vertical stratification of plant structure, while the second one reflects the horizontal variation of plant structure as the aggregate of data on tree height and crown projection area (Supplementary Table 27). For the grassland system, we used Rao's $Q$ and Moran's $I$ (the latter multiplied by −1 to reflect increasing heterogeneity) of height distribution directly as metrics of vertical stratification and horizontal variation.

**Path models**. We used path analysis[71] to assess the direct (paths from plant diversity to arthropod richness) and indirect (paths via arthropod abundance) effects of taxonomic, functional, and structural diversity of the plant communities on arthropod richness. As potential predictors, we considered plant species richness (planted or sown number of species per plot), plant functional diversity (Rao's $Q$ of plant traits), plant trait composition (based on CWMs), vertical stratification and horizontal variation of plant structure (based on Rao's $Q$ and Moran's $I$ of plant height and, for the forest system, crown projection area), and plant biomass. We fitted individual models for overall arthropod species richness, as well as for the species richness of herbivores, predators, and parasitoids.

The initial models included the most relevant pathways derived from theoretical assumptions and correlations among the plant-based predictors (Supplementary Fig. 4). We assumed that plant species richness, as the experimental treatment variable, influences plant biomass[7], functional leaf trait diversity[72], and structural diversity[41]. Functional trait composition (PC2) was only marginally significantly related to plant species richness in the grassland experiment but not in the forest experiment. We therefore only considered this path in the grassland models. Moreover, we assumed that all plant-based predictors can directly influence arthropod abundance and species richness[16,22,25]. Finally, we expected arthropod abundance to influence arthropod species richness (e.g., more individuals hypothesis[37,73]). We additionally tested for significant residual covariances between the plant-based predictors (see Supplementary Fig. 4), as the different components of plant diversity might not be completely independent. We sequentially dropped non-informative pathways and covariances, if their removal resulted in a reduction of the AICc of the models[16,71]. The final models were those that minimized AICc values and included 0 in the 95% confidence interval of the root mean square error of approximation. We tested the robustness of the results by calculating bootstrapped $P$ values based on 1000 bootstrap draws[71]. Arthropod data, plant biomass, plant species richness, and functional diversity were log-transformed for the analyses.

Based on this path modeling approach, we additionally tested two alternative path model variants. The first variant used the same initial models as described above, except for a residual covariance term between arthropod species richness and abundance rather than a directional pathway between the two. We simplified models as described above and compared the resulting AICc values to those of the final models of our initial approach. We considered the model variant with the

lowest AICc as better supported when differences in AICc were >2, otherwise both model variants and their underlying hypotheses (directionality of abundance-richness relationships vs. abundance-richness covariance) were considered to be equally likely[74]. In a second variant, we based the path models on rarefied arthropod species richness (based on the minimum number of individuals per plot for each higher trophic level) to test how our interpretation of plant diversity effects on arthropod species richness changes after factoring out the potentially important influence of arthropod abundance (note: rarefaction was not possible for parasitoids in the forest experiment because the lowest number of individuals per plot was 1). Again, we used the same general model structure and simplification procedure as described above. However, because arthropod abundance was factored out by rarefaction, we did not include abundance and the corresponding pathways via abundance in these models.

All analyses were conducted in R 3.3.1 (www.r-project.org) with the packages vegan, FD, VoxR, and lavaan.

**Reporting Summary**. Further information on experimental design is available in the Nature Research Reporting Summary linked to this article.

## Data availability
Data used in the analyses is available on the data repository of the German Centre of Integrative Biodiversity Research (iDiv) at https://doi.org/10.25829/idiv.295-17-1066. A reporting summary for this article is available as a Supplementary Information file. The source data underlying Figs. 1–3 and Supplementary Fig. 4 are provided as a Source Data file.

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

## Acknowledgements

We thank Xuefei Yang, Chen Lin, Sabine Both, Xiaojuan Liu, Yang Bo, Keping Ma and all members of the BEF-China consortium who coordinated and helped with the establishment and maintenance of the experiment. We thank the technical staff of the Jena Experiment consortium for their work in maintaining the field site and many student helpers for weeding of the experimental plots and support during measurements. We acknowledge Eric Anton, Theo Blick, Frank Creutzburg, Günter Köhler, Gerlinde Kratzsch, Michael Meyer, Christoph Muster, and Oliver Wiche for their tremendous work of grassland arthropod identification. We are grateful to Stefan Michalski for providing phylogenetic plant data. We are indebted to our many helpers who contributed to the sampling of arthropods and plant data in the experiments. We gratefully acknowledge funding by the German Research Foundation (DFG FOR 891/1-3), the Sino-German Centre for Research Promotion (GZ 524, 592, 698, 699, 785, and 1020), and the National Natural Science Foundation of China (NSFC 30710103907 and 30930005). The Jena Experiment is funded by the German Research Foundation (DFG FOR 1451) with additional support from the Friedrich Schiller University Jena, the Max Planck Society, and the Swiss National Science Foundation (SNF). A.S., H.B., and N.E. acknowledge support by the German Centre for Integrative Biodiversity Research (iDiv) Halle-Jena-Leipzig (DFG FZT 118). N.E. acknowledges funding by the European

Research Council (ERC Starting Grant; grant agreement no. 677232). We acknowledge the financial support of the Open Access PublicationFund of the Martin-Luther-University Halle-Wittenberg. We thank theBiodiversity Informatics Unit of iDiv for providingsupport with the data quality checks and uploading to the data repository.

## Author contributions

N.E. conceived the idea for the manuscript; N.E. and H.B. coordinated the Jena and BEF-China experiment, respectively; A.S., N.E., M.K., A.E. and M.S. developed the study; A.S., A.E., M.S., N.B., W.H., A.-M.K., C.R., G.v.O., A.W., C.W., W.W., H.B. and N.E. designed research; A.S., A.E., M.K., M.S., C.G.-S., D.B., W.D., A.F., F.F., L.H., C.R., J.S. and J.Z. collected and/or contributed data; A.S. conducted the statistical analyses and wrote the manuscript, with input from all coauthors.

## Additional information

**Competing interests:** The authors declare no competing interests.

