## [Peer Review File · Nature Communications]

Reviewers' Comments:

Reviewer #1:

Remarks to the Author:

In their study Schuldt et al. investigate the effects of various components of plant diversity (taxonomic, functional and structural diversity) on the diversity of higher trophic levels (herbivores, predators, parasitoids) in the context of two large-scale biodiversity experiments. The authors find that the positive effects of plant diversity on the diversity of higher trophic levels were mainly mediated by the structural and functional diversity of the plant communities, but the relative importance of these components differed between trophic levels and ecosystems. Moreover, the structural equation models suggest that positive effects of plant diversity on consumer diversity are indirectly mediated by increased consumer abundance.

The analysis is based on an impressive dataset that has been compiled from two large-scale and long-term biodiversity experiments: one in temperate grasslands (Jena Experiment) and one in subtropical forests (BEF-China forest experiment).

The novelty of the work lies in the authors' effort to integrate not only the direct effect of taxonomic plant diversity on consumer diversity but also the indirect effects via functional AND structural diversity of plant communities. In this aspect, the study goes beyond the results of previous work that mainly focussed on taxonomic diversity (e.g., Scherber et al. 2010; Schuldt et al. 2015, Castagneyrol & Jactel 2012).

Moreover, the authors emphasize the finding that effects of plant diversity on consumer diversity are mediated by consumer abundance, which is in line with the 'more individuals hypothesis' (Storch et al. 2018; i.e., more diverse plant communities can harbour more viable consumer populations and alongside a higher consumer diversity). The authors argue that this could be an important pathway that links plant diversity to consumer diversity and may be relevant for improving predictions of biodiversity change in response to global environmental change.

I found the general ideas presented in the manuscript (the role of different components of plant diversity and the importance of different bottom-up pathways) very interesting and timely. Also studying the indirect effect of plant diversity on consumer diversity via consumer abundance in the context of a biodiversity experiment is timely. This is particularly true in the light of recent work that reported strong declines in insect abundance over the last decades (Hallman et al. 2017). The causes of these declines and the consequences for biodiversity still remain unclear (as acknowledged by the authors). Therefore, the manuscript is likely of interest to others in the community and to a wider audience.

I have, however several methodological concerns that the authors should address:

A) The abundance and species richness of consumers are the two primary response variables in the paper and the main claims are all based on patterns related to these two variables. With this in mind, I was surprised that the authors provide no justification for removing singletons prior to calculation of consumer species richness (and abundance). So my main concern is to what extent are the results influenced by the decision to remove singletons from the dataset. What was the basis for the decision to remove singletons? The point is also important, because removing singletons will not only affect consumer richness, but also abundance. So ultimately it remains unclear, how robust the main claims of the paper related to the mediating effect of consumer abundance on consumer richness actually are. At a minimum the authors should show in a supplementary analysis that they obtain the same results regardless of the decision to remove singletons or not. Probably a better way to overcome the

arbitrary decision to remove singletons would be to either use rarefied species richness (based on the rarefy function in the R package *vegan*) or to use an asymptotic diversity estimator (e.g., the Chao1 estimator as implemented through the function *ChaoRichness* in the R package *iNEXT*; Hsieh et al. 2016). These methods do not only use all information that is available in the species abundance data, but they are also likely to be more robust given that in the forest experiment the sampling intensity (i.e., the number of trees sampled) differed between plots (Lines 390-394). It is true that the authors tried to account for this, but taking the residuals from a regression of consumer richness against number of tree individuals sampled in a subsequent analysis may not completely accounted for differences in sampling intensity. Thus, I strongly recommend using one of the alternative methods.

B) One of the two main findings of the paper is that plant diversity is linked to consumer diversity via consumer abundance, which the authors interpret as support for the 'more individuals hypothesis' (Storch et al. 2018). My main concern regarding this claim is that the authors did not assess any alternative hypotheses. In fact, in their paper Storch et al. (2018) present two potential scenarios of how consumer diversity is related to consumer abundance and productivity (Fig. 4 in the cited paper). The first scenario assumes an indirect effect of producers on the diversity of consumers via consumer abundance. The second scenario assumes that producers have direct effects on consumer abundance and richness, and that abundance and richness are themselves correlated with each other due to processes such as extinction (, speciation) and energy use. This second model is best represented by a residual covariance term between consumer abundance and richness, because the processes that underlie any correlation between the two variables would then be unknown (and unmeasured). Therefore, the claim that producers affect consumer diversity via abundance needs to be substantiated by explicitly contrasting these two alternative hypotheses (i.e., a directed effect of abundance on diversity versus residual covariance between abundance and diversity). Two nice examples for such comparisons of the directionality of effects in structural equation models can be found in Scherber et al. (2010) and Sandom et al. (2013).

C) The authors devote a large part of the results section to explaining how the strength of the biodiversity effects of the two main components (structural or functional diversity) differs between trophic levels (Lines 166-196). I have the impression that it would be helpful to add a third figure to the main text (e.g., in the form of a bar plot), which summarizes the direct and indirect effects of plant diversity on higher trophic levels that are mediated by each of the diversity components (function & structure).

This figure could aid the results and discussion section and may help readers to follow the line of argument developed by the authors, regarding the contributions of structural or functional attributes to the effect of plant diversity on the diversity of higher trophic levels. As an example, the bar plot could illustrate 1.) the sum of the indirect plant diversity effects that are mediated by structural attributes, 2.) the sum of the indirect plant diversity effects that are mediated by functional attributes, and 3.) the direct effect of taxonomic diversity (and biomass?). These summed effects could then be shown for all arthropods and for each trophic level separately to emphasize the differences in direction and magnitude of the indirect effects.

Similarly, the figure could also contrast the direct effect of plant diversity on consumer diversity and the indirect abundance-mediated effect of plant diversity on consumer diversity. This is particularly important, as one of the main claims in the paper is that plant diversity mainly affects consumer diversity via consumer abundance. Yet, comparing the direct and indirect effects in many cases suggests that the direct effects are at least as strong or stronger than the indirect abundance-mediated effects (Figs. 1 and 2). For instance, in Figure 1b the direct effect of functional diversity on arthropod species richness is $0.69 \times 0.24 = 0.166$, whereas the indirect abundance-mediated effect is $0.69 \times 0.2 \times 0.64 = 0.088$, which is roughly 50% of the direct effect.

Of course, there are many potential ways of grouping the effects and the grouping will depend on which aspects of the data the authors would like to highlight. But I think that a figure, as proposed above, may provide some additional insights about whether the importance of the components differs between trophic levels and/or ecosystems that would otherwise be hard to capture for the non-expert reader. In addition, the figure could be useful to guide those readers that are less familiar with structural equation modelling (given the variation in complexity of the final models presented in Figure 2). One example for this approach (even though not exactly) can be found in Allan et al. (2015).

D.) More care is needed when generalizing the results from two ecosystems (forest and grassland) without having replicated experiments in each ecosystem type. This is particularly important, because the data from each experiment are treated differently regarding (i) the inclusion or exclusion of particular consumer taxa (Lines 366, 367 & 381), (ii) the treatment of structural diversity metrics prior to analysis (Lines 509-516), and (iii) the inclusion of residual covariance terms in the SEMs (Line 536). All of these aspects of data treatment are likely to cause systematic differences, that may have nothing to do with underlying system-specific differences based on underlying ecological principles.

E.) Line 537: The authors used AIC for model selection, even though the sample sizes in the analyses are relatively small ($n = 46$ for the forest experiment and $n = 92$ for the grassland experiment). In this situation AICc is better suited for model selection than AIC (Johnson & Omland 2004), as AIC tends to favour over-fitted models when sample sizes are small (Burnham et al. 2011). This also seems to be the case in the present study, as many marginally non-significant paths are retained in the final path models, that may have been excluded when applying more conservative selection criteria (Figs, 1 and 2).

Specific comments:

Discussion

Line 208: It is better to avoid statements like 'for the first time'.

Lines 272-276: Here the authors claim that of the many potential pathways, indirect effects via arthropod abundance were particularly important. It is unclear what the authors mean by important. Is it the strength of the indirect relationships or the fact that the effect of arthropod abundance on richness consistently appeared throughout the models? Figures 1 and 2 suggest that the second alternative more closely reflects the results of the SEMs.

Lines 286-290: I very much like this example of how the results could inform conservation.

Methods:

Lines 366: What were the criteria for classifying taxa as 'sufficiently sampled' in the forest experiment?

Lines 367 & 381: While the taxonomic groups considered in the two experiments do partly overlap, some of the groups were only considered in one of the experiments (e.g., Blattodea, Mantodea, Psocoptera and Lepidoptera were only considered in the forest experiment, whereas Isopoda, Myriapoda and Chilopoda were only considered in the grassland experiment). To what extent can the differences in the patterns between the two experiments/ecosystems be explained by differences in the composition of the analysed taxonomic groups?

Lines 390-394: The sampling intensity (i.e., the number of trees sampled) differed between plots in

the forest experiment. As mentioned earlier, it would be valuable if the authors could show by using alternative approaches (asymptotic richness estimators or rarefied species richness) that their results are robust. This is particularly important, because the abundance and species richness of consumers are the two primary response variables in the paper and the main claims are based on patterns related to these two variables.

Line 430: This sentence either contains a repetition, or there might be a full stop missing after 'Leaf toughness and silica concentration were assessed.'

Lines 509-516: Here the authors treat the structural diversity variables of the two experiments differently, which raises the question to what extent differences in the importance of structural diversity between ecosystem are driven by differences in the treatment of the respective variables. Do the authors obtain similar results regarding the effects of the structural variables in the grassland experiment, if the two variables in this grassland experiment are also subject to PCA prior to the analyses?

Line 536: Including residual covariance terms into path models only to improve model fit is not a well-justified argument. Residual covariance terms are included into structural equation models to account for common (but potentially unmeasured) sources of variance among the variables in a model. That is, covariance terms should be included if there is some a priori information about a potential correlation between two variables for which the causal structure/direction is unclear or unknown. Apart from that, I agree with the authors that the models should certainly contain residual covariance terms. I was wondering, however, why these terms were only included in the grassland models and why only certain covariance terms were considered. Were some covariance terms not considered because, the explanatory variables were extracted from PCAs? Did the authors exclude certain covariance terms based on AIC as for the directed paths? The criteria for including or excluding covariance terms should be described more clearly.

References:

Allan et al. (2015). Land use intensification alters ecosystem multifunctionality via loss of biodiversity and changes to functional composition. *Ecology Letters*, 18 (8), 834–843. doi:10.1111/ele.12469

Burnham, K. P.; Anderson, D. R.; Huyvaert, K. P. (2011). AIC model selection and multimodel inference in behavioral ecology. *Behavioral Ecology and Sociobiology*, 65, 23–35. doi:10.1007/s00265-010-1029-6

Castagneyrol, B., & Jactel, H. (2012). Unraveling plant–animal diversity relationships: a meta-regression analysis. *Ecology*, 93 (9), 2115–2124. doi:10.1890/11-1300.1

Hallmann et al. (2017). More than 75 percent decline over 27 years in total flying insect biomass in protected areas. *PLOS ONE*, 12 (10), e0185809. doi:10.1371/journal.pone.0185809

Hsieh, T. C., Ma, K. H., & Chao, A. (2016). iNEXT: an R package for rarefaction and extrapolation of species diversity (Hill numbers). *Methods in Ecology and Evolution*, 7 (12), 1451–1456. doi:10.1111/2041-210x.12613

Johnson, J. B., & Omland, K. S. (2004). Model selection in ecology and evolution. *Trends in Ecology & Evolution*, 19 (2), 101–108. doi:10.1016/j.tree.2003.10.013

Sandom et al. (2013). Mammal predator and prey species richness are strongly linked at macroscales. *Ecology*, 94 (5), 1112–1122. doi:10.1890/12-1342.1

Scherber et al. (2010). Bottom-up effects of plant diversity on multitrophic interactions in a biodiversity experiment. *Nature*, 468 (7323), 553–556. doi:10.1038/nature09492

Schuldt et al. (2015). Multitrophic diversity in a biodiverse forest is highly nonlinear across spatial scales. *Nature Communications*, 6 (1). doi:10.1038/ncomms10169

Storch, D., Bohdalková, E., & Okie, J. (2018). The more-individuals hypothesis revisited: the role of community abundance in species richness regulation and the productivity-diversity relationship. *Ecology Letters*, 21 (6), 920–937. doi:10.1111/ele.12941

Reviewer #2:

Remarks to the Author:

The Authors tackle a highly interesting question, i.e. the role of structural and functional plant diversity in shaping the higher trophic levels (arthropod communities). To answer this, they measured specific plant traits and collected a high number of arthropod specimens, in experimental plots set in two different ecosystems, a subtropical forest in China and a grassland in Germany, they calculated functional and structural diversity indices, and they used path analysis in order to test the relationships between these indices and arthropod abundance in the plots of the examined ecosystems.

I love the concept of this study, more specifically:

1. The use of two different ecosystems is important for detecting/describing trade-offs in the functional roles of plant traits, according to local conditions.
2. I appreciate their extensive fieldwork samplings; I understand how painful this process can be, and I acknowledge their effort (especially given all the methodological implications included in it).
3. Sampling multiple trophic levels and functional guilds can offer a holistic view of the community.
4. Structural plant diversity as a predictor of arthropod diversity is indeed an overlooked factor.
5. Sampling multiple plant traits can also be painful, yet it allows a thorough exploration of interspecific relationships in study communities.

However, I'm afraid that I do not agree with the Authors claim that, with their current analytical approach, they really disentangle the drivers of arthropod diversity regarding plants.

For example, after reading the Discussion, I was left wondering if leaf trait diversity is really insignificant. But then again, why would a leaf trait diversity index be significant in describing one or two ecosystems? What would be the practical significance of this result, and how could it be used for restoration or conservation to tackle specific problems regarding diversity of the different trophic levels? On the other hand, the evaluation of specific traits could really guide us to conclusions useful for designing management schemes, or even help sketch the phenotypic profile of the functionally important species in the two ecosystems studied. These are topics that I missed from the Discussion, and I think that they cannot be answered with the current analytical scheme. Finding relationships between diversity indices, (a) perhaps is not novel enough, and (b) can hardly provide details on the mechanistic dimension of the relationships between plant and animals (it would probably do so on a macroecological scale, but here we have only two experimental study areas).

Again, as for the leaf traits, maybe there are indeed specific ones that matter for (i.e. predict the diversity of) the different trophic levels or functional guilds. Have you tried testing the mean values of

leaf traits against arthropod diversity? Functional diversity indices are very useful, but I am afraid that in this particular case, alternative analyses are needed in order to actually disentangle the role of the components of plant functional diversity that predict arthropod diversity in the communities studied. I really think that this is an amazing dataset, and most importantly, it allows for the application of many different analytical methods. There is, for example, the possibility of using Multivariate-response GLMs, and test each plant trait (or multiple predictors) against the entire community of the arthropods captured in each area.

To return explain this lack of significant relationships, the Authors argue that there might be more traits that have not been considered in this study, however, I doubt this.

Have the Authors tested this hypothesis?

Moreover, what I felt was missing from this approach is the role of phylogenetic diversity of both the plant and the animal assemblage. Multispecies trait-based analyses benefit from assessing the role of phylogeny in predicting the distribution of traits among the members of the community, because type I errors due to phylogenetic similarity (phylogenetic pseudoreplication) are avoided.

The way that the questions of this analysis are currently structured, phylogenetically-corrected models (PGLS, PGLMM, etc.) are probably not applicable. However, is the phylogenetic dissimilarity of species within plots, a factor linked to arthropod species distribution? I understand that the vegetation plots are experimental (i.e. planted, not entirely natural); still, similarities in plant traits or metabolites (which are related to phylogeny and are not measured here) could potentially drive the composition of arthropod communities, as suggested by the Authors in L246-247.

Pairwise phylogenetic dissimilarity matrices are easy to generate both for plants (e.g. Phylomatic software) and for the animals --especially given that DNA barcoding has been used for verifying taxonomic identifications of the arthropod specimens.

To sum up, although I love the concept of this work, I have two main concerns: First, the use of functional diversity indices, instead of the traits measured, and second, the fact that phylogeny is omitted from the analyses.

In other respects, this is a very well written and structured manuscript, with excellent use of the English language, presenting methodology in a clear and comprehensive way. My specific, line-by-line comments are minor:

INTRODUCTION

- The Introduction is excellent, presenting a thorough review of the relevant literature, and clearly explaining the aims of the study.

L61-62: "Mixed support" is not a very clear term.

L108-111: I am not sure that I entirely understand the Authors' distinction between the direct and the indirect effects of plant diversity to arthropod diversity at this point of the text. I would thus recommend rephrasing.

RESULTS

- Perhaps change some decimal separators in the text, from comma to dot?
- Although I understand the style of presenting results of path analysis models, I am afraid that the figures are way too complex. Perhaps I would remove the clipart, in order to make the figures less dense and crowded to the eye. Also, please mind that the current brown boxes cannot be distinguished from the pink ones by a person with color-blind vision.

L767-768: Grammar

L768: Can you explain "endogenous" variables?

Fig. 1: The explanation of the c) section of the figure is not clearly noted (it starts in L766?).

METHODS

- Were there any pollinators collected with the suction method? Are there mutualists in the dataset --I read about parasitoids to Hymenoptera but are there parasitoids to herbivores?
- I guess it is already included in the Supp. Material (although I couldn't have access to it), but I would like to see a list of the species sampled, and their classification into groups (predators, parasitoids, etc.).

L430: Please delete "Leaf toughness and silica concentration were assessed"

L519-520: Please define direct and indirect.

Reviewer #3:

Remarks to the Author:

Schultz et al. Multiple plant diversity components drive consumer communities across ecosystems.

Brief summary

This manuscript evaluates the relative importance of several mechanisms by which plant diversity affects arthropod communities in two different ecosystems, a temperate grassland and a subtropical forest. Specifically, the authors use path analyses to compare the effects of plant species richness, biomass, trait composition, functional diversity, and structural diversity (horizontal variability and vertical stratification) on total arthropod abundance and total arthropod species richness, as well as for three trophic levels separately (herbivores, predators, and parasitoids). They conclude that plant species richness has a consistent and positive effect on arthropod species richness, which is mediated by greater plant structural and functional diversity, however the specific mechanisms differed between ecosystems and trophic levels.

In general, the manuscript is well-written and would be relevant to a broad audience. The effects of plant diversity are of significant interest in ecology, however, this field has been limited to one or two trophic levels, and rarely have studies documented the separate trophic groups as done here. The described dataset is substantial in terms of the arthropod sampling and identification, breadth of traits measured, and structural components included in the analyses. However, the generalities that can be taken from the comparison of the grassland and forest ecosystems are limited due to the low sample size within each system, difference in climate, and general differences in experimental design (and perhaps more could be said to this end). Regardless, the combined dataset is impressive and highlights the complexities of biodiversity effects from a community-perspective.

In measuring plant functional traits and diversity, the complete sampling design and resolution of the data is somewhat unclear. Were plant trait data taken for the same plots and individual trees that arthropods were sampled? Based on lines 422-428, trait values for trees are based on 5 individuals per species, while in the grassland traits were measured from 3 individuals per plot? Lines 422-423 and lines 428-429 are conflicting. If trait data are available at the individual plant or plot-level, it seems that these data should be used to relate directly to arthropod data from the corresponding individuals/plots, rather than species-level means which may eliminate variance that is highly relevant to the sampled communities.

Overall the statistical approach is solid, although the sample size (plots, see above) could be argued to be low for the complexity of the models being tested. In particular, the forest analysis is based upon a sample size of 42 (plots); is this sufficient to assess the inter-relationships among 9 variables? I'm not sure. In addition, the results are somewhat overwhelming due to the number of path diagrams (8) and separate tests

for each trophic group as well as all arthropods together in both the forest and grassland. While I think the results from each of these individual tests are interesting, it is hard to digest with the short format of the article, where there is not enough space to discuss in depth the different hypotheses related to each test. Moreover, to consider each trophic level separately ignores any potential indirect effects of plant diversity mediated by co-occurring arthropods, like density- or diversity-dependent responses of natural enemies to herbivores. This is likely important, and what would be interesting is to compare the strength of such herbivore-mediated effects to the direct effects of plant diversity, although this of course adds complexity.

I am also concerned that In addition, the relationship arthropod species richness and abundance is directional but it could be argued that this should be treated as a covariance. It is also not clear which other covariances were included in the path models. The path between plant species richness and trait composition is omitted from the analyses, with the exception of plant species richness and trait composition PC2 in the grassland only. Is it accurate to say tree biomass causes structural diversity, or that these covary?

Finally, there is some context for the novelty of the results presented here that isn't addressed. For example, there is a large literature on the Enemies Hypothesis showing the linkage between plant diversity and predator abundance and diversity. And its well understood that structural complexity affects predator abundance / diversity by reducing intra-guild predation. Some of the relevant literature on these topics is cited, but the text doesn't really make clear that some of the results presented here are anticipated or previously demonstrated by this literature.

Lines 108-113: Do increases in abundance cause increases in richness? Clearly they are tied to one another

Supplement

Table 2 – Shouldn't PC1 inherently explain a greater proportion of variance than PC2?

Specific comments:

Line 85-87: Repetitive to previous sentence

Line 90-92: Suggest dropping "consequences of such changes" to "consequences" and making a distinction between trophic groups. I am not sure this sentence describes an indirect effect of plant diversity, it is somewhat unclear. Is it meant that variation in plant diversity affects arthropod diversity through arthropod abundances? In the case of herbivores, would this not be a direct effect of plant diversity, whereas for a predator or parasitoid it could be indirect if they are responding to herbivores (or direct if they are responding to plant structural complexity)?

Line 98-100: Is a result general if detected in 2 vs 1 experiment? What about within-system variation? As stated in the introduction, there are mixed results from biodiversity experiments regarding plant diversity effects on higher trophic levels so I find it difficult to make this comparison between grassland and forest.

Line 102-106: Too much in parentheses. The explanations of the calculations could come later. Hypothesis "related to microclimate and habitat space" introduced earlier and could be cut here. The specific components of functional diversity could be fleshed out more.

Line 122-135: combine paragraphs?

Line 133-135: Wording unclear - "correlations between abundance and/or richness values across higher trophic levels". Are these correlations between arthropod abundance and arthropod richness? Or between plants and arthropods?

Line 139: "either directly or indirectly via effects..."

Line 144 & 146: "for the log-log relationship" unclear

Line 146: "tended" to "showed a marginal relationship" or similar

Line 162-164: 171-175: Not clear what aspect of arthropods has an effect on, abundance?

Line 202: "between" not "among"

Line 231-234: States there are differences in sampling methods but then says they are the same.

Line 234-236: What is the argument for restricting the analysis?

Line 249-251: This implies that leaf trait effects...not tree diversity

Line 310-314: If 46 plots were used, why mention 64? Why is there lower replication at higher levels of diversity?

Line 315-319: Less diverse = monocultures? The explanation of how plots were selected is unclear. What is meant by non-overlapping fractions? Why are the 24-species plots "additional"? Do the 24-species plots contain species not present in any of the other plots?

Line 403: what does 6x6 and 12x12 mean? Do plots differ in the number of trees?

Line 430-431: "Leaf toughness and silica concentration were assessed Leaf toughness...."

Line 431-434: Mention of mesocosm plants is a surprise

Line 455-458: Include this information earlier in main text as well

Reviewers' comments:

Reviewer #1 (Remarks to the Author):

In their study Schuldt et al. investigate the effects of various components of plant diversity (taxonomic, functional and structural diversity) on the diversity of higher trophic levels (herbivores, predators, parasitoids) in the context of two large-scale biodiversity experiments. The authors find that the positive effects of plant diversity on the diversity of higher trophic levels were mainly mediated by the structural and functional diversity of the plant communities, but the relative importance of these components differed between trophic levels and ecosystems. Moreover, the structural equation models suggest that positive effects of plant diversity on consumer diversity are indirectly mediated by increased consumer abundance.

The analysis is based on an impressive dataset that has been compiled from two large-scale and long-term biodiversity experiments: one in temperate grasslands (Jena Experiment) and one in subtropical forests (BEF-China forest experiment).

The novelty of the work lies in the authors' effort to integrate not only the direct effect of taxonomic plant diversity on consumer diversity but also the indirect effects via functional AND structural diversity of plant communities. In this aspect, the study goes beyond the results of previous work that mainly focussed on taxonomic diversity (e.g., Scherber et al. 2010; Schuldt et al. 2015, Castagneyrol & Jactel 2012).

Moreover, the authors emphasize the finding that effects of plant diversity on consumer diversity are mediated by consumer abundance, which is in line with the 'more individuals hypothesis' (Storch et al. 2018; i.e., more diverse plant communities can harbour more viable consumer populations and alongside a higher consumer diversity). The authors argue that this could be an important pathway that links plant diversity to consumer diversity and may be relevant for improving predictions of biodiversity change in response to global environmental change.

I found the general ideas presented in the manuscript (the role of different components of plant diversity and the importance of different bottom-up pathways) very interesting and timely. Also studying the indirect effect of plant diversity on consumer diversity via consumer abundance in the context of a biodiversity experiment is timely. This is particularly true in the light of recent work that reported strong declines in insect abundance over the last decades (Hallman et al. 2017). The causes of these declines and the consequences for biodiversity still remain unclear (as acknowledged by the authors). Therefore, the manuscript is likely of interest to others in the community and to a wider audience.

-Thank you for your time and effort and the many helpful suggestions for improvement!

I have, however several methodological concerns that the authors should address:

A) The abundance and species richness of consumers are the two primary response variables in the paper and the main claims are all based on patterns related to these two variables. With this in mind, I was surprised that the authors provide no justification for removing singletons prior to calculation of consumer species richness (and abundance). So my main concern is to what extent are the results influenced by the decision to remove singletons from the dataset.

What was the basis for the decision to remove singletons? The point is also important, because removing singletons will not only affect consumer richness, but also abundance. So ultimately it remains unclear, how robust the main claims of the paper related to the mediating effect of consumer abundance on consumer richness actually are. At a minimum the authors should show in a supplementary analysis that they obtain the same results regardless of the decision to remove singletons or not. Probably a better way to overcome the arbitrary decision to remove singletons would be to either use rarefied species richness (based on the rarefy function in the R package *vegan*) or to use an asymptotic diversity estimator (e.g., the Chao1 estimator as implemented through the function *ChaoRichness* in the R package *iNEXT*; Hsieh et al. 2016). These methods do not only use all information that is available in the species abundance data, but they are also likely to be more robust given that in the forest experiment the sampling intensity (i.e., the number of trees sampled) differed between plots (Lines 390-394). It is true that the authors tried to account for this, but taking the residuals from a regression of consumer richness against number of tree individuals sampled in a subsequent analysis may not completely accounted for differences in sampling intensity. Thus, I strongly recommend using one of the alternative methods.

-We thank the Reviewer for raising this important point. We removed singletons to make the data set more robust, because species recorded with just one individual in the whole data set might be vagrants that are not really associated with the study systems. Following the Reviewer's suggestion, we again checked the data with and without singletons and found that species richness in both data sets was highly correlated (Pearson's $r > 0.97$, $P < 0.001$ in all cases and for all trophic levels), meaning that removal of singletons did not change the structure of our data. We have added this information to the Methods (L447-450). We now also provide the results of an alternative analysis based on rarefied data (individual-based rarefaction with the R-package *vegan*), as suggested by the Reviewer (Supplementary Tables 18-24). These results show that when factoring out the effects of differences in the abundance of arthropods on arthropod species richness, many of the indirect and direct effects of plant diversity on arthropod species richness disappeared. This underlines the role of arthropod abundance in modifying the relationships between plant and arthropod diversity, which is also the reason why we chose the path models that explicitly included pathways from plant to arthropod diversity via arthropod abundance as our main analysis. Rarefaction (or extrapolation) standardizes arthropod species richness to an equal number of arthropod individuals per study plot, which precludes a quantification of plant diversity effects on arthropod diversity via arthropod abundances. However, this was one of the aims of our study, highlighting that we need to take changes in arthropod abundances into account when trying to understand the mechanisms underlying the effects of plant diversity on arthropod diversity (and thus highlights the importance of considering abundances in the context of functional biodiversity research). We describe the path models based on rarefied species richness in the Methods (L630-638) and added the outcome of these analyses to the Results (L184-188): *“Models based on rarefied arthropod species richness showed that when factoring out arthropod abundance, many of the indirect and direct effects of plant diversity on arthropod species richness disappeared (Supplementary Tables 18-24), underpinning the role of arthropod abundance changes in modifying the relationships between plant and arthropod diversity.”*

B) One of the two main findings of the paper is that plant diversity is linked to consumer diversity via consumer abundance, which the authors interpret as support for the ‘more individuals hypothesis’ (Storch et al. 2018). My main concern regarding this claim is that the authors did not assess any alternative hypotheses. In fact, in their paper Storch et al. (2018)

present two potential scenarios of how consumer diversity is related to consumer abundance and productivity (Fig. 4 in the cited paper). The first scenario assumes an indirect effect of producers on the diversity of consumers via consumer abundance. The second scenario assumes that producers have direct effects on consumer abundance and richness, and that abundance and richness are themselves correlated with each other due to processes such as extinction (, speciation) and energy use. This second model is best represented by a residual covariance term between consumer abundance and richness, because the processes that underlie any correlation between the two variables would then be unknown (and unmeasured). Therefore, the claim that producers affect consumer diversity via abundance needs to be substantiated by explicitly contrasting these two alternative hypotheses (i.e., a directed effect of abundance on diversity versus residual covariance between abundance and diversity). Two nice examples for such comparisons of the directionality of effects in structural equation models can be found in Scherber et al. (2010) and Sandom et al. (2013).

-We agree with the Reviewer and have added path model analyses with a covariance term between arthropod abundance and arthropod species richness to our manuscript (Methods: L622-630, Results L177-184). We also explain the theoretical expectations for these alternative models in the Introduction (L119-122):

“We also tested the alternative hypothesis of reciprocal interactions between arthropod species richness and abundance³⁸, which might be better reflected by residual covariance terms than by a directional pathway in the path models.”

Our analyses show that models based on direct paths from arthropod abundance to arthropod species richness are generally better supported in both study systems (lower AICc values) than models fitted with an abundance-richness covariance. Only in the case of forest and grassland herbivores were both types of models equally well supported (similar AICc values; L177-184). Overall, this lend further support to our assumption that the patterns reported here are driven by mechanisms described by the more individual hypothesis.

C) The authors devote a large part of the results section to explaining how the strength of the biodiversity effects of the two main components (structural or functional diversity) differs between trophic levels (Lines 166-196). I have the impression that it would be helpful to add a third figure to the main text (e.g., in the form of a bar plot), which summarizes the direct and indirect effects of plant diversity on higher trophic levels that are mediated by each of the diversity components (function & structure). This figure could aid the results and discussion section and may help readers to follow the line of argument developed by the authors, regarding the contributions of structural or functional attributes to the effect of plant diversity on the diversity of higher trophic levels. As an example, the bar plot could illustrate 1.) the sum of the indirect plant diversity effects that are mediated by structural attributes, 2.) the sum of the indirect plant diversity effects that are mediated by functional attributes, and 3.) the direct effect of taxonomic diversity (and biomass?). These summed effects could then be shown for all arthropods and for each trophic level separately to emphasize the differences in direction and magnitude of the indirect effects.

Similarly, the figure could also contrast the direct effect of plant diversity on consumer diversity and the indirect abundance-mediated effect of plant diversity on consumer diversity. This is particularly important, as one of the main claims in the paper is that plant diversity mainly affects consumer diversity via consumer abundance. Yet, comparing the direct and indirect effects in many cases suggests that the direct effects are at least as strong or stronger than the indirect abundance-mediated effects (Figs. 1 and 2). For instance, in Figure 1b the direct effect of functional diversity on arthropod species richness is $0.69 \times 0.24 = 0.166$,

whereas the indirect abundance-mediated effect is $0.69 \times 0.2 \times 0.64 = 0.088$, which is roughly 50% of the direct effect.

Of course, there are many potential ways of grouping the effects and the grouping will depend on which aspects of the data the authors would like to highlight. But I think that a figure, as proposed above, may provide some additional insights about whether the importance of the components differs between trophic levels and/or ecosystems that would otherwise be hard to capture for the non-expert reader. In addition, the figure could be useful to guide those readers that are less familiar with structural equation modelling (given the variation in complexity of the final models presented in Figure 2). One example for this approach (even though not exactly) can be found in Allan et al. (2015).

-Thank you very much for this suggestion, which helps to present our main results more clearly. As recommended, we now provide a third figure (Figure 3) showing the overall effects of plant taxonomic, functional, and structural diversity on arthropod species richness, where we differentiate between direct effects of the different components of plant diversity on arthropod species richness (direct paths between plants and arthropod richness) and indirect effects via arthropod abundance. This figure helps to summarize our main findings and shows that in many cases indirect effects via arthropod abundance can be as important as, or even more important than, the direct effects of plant diversity on arthropod species richness.

D.) More care is needed when generalizing the results from two ecosystems (forest and grassland) without having replicated experiments in each ecosystem type. This is particularly important, because the data from each experiment are treated differently regarding (i) the inclusion or exclusion of particular consumer taxa (Lines 366, 367 & 381), (ii) the treatment of structural diversity metrics prior to analysis (Lines 509-516), and (iii) the inclusion of residual covariance terms in the SEMs (Line 536). All of these aspects of data treatment are likely to cause systematic differences, that may have nothing to do with underlying system-specific differences based on underlying ecological principles.

-We agree and have added several notes and explanations in the text that more clearly explain these issues (L249-251, L613-617). In general, we consider these differences to be due primarily to differences between the two experiments in the structure and type of ecosystem. Please see our replies the comments below for details on these issues.

E.) Line 537: The authors used AIC for model selection, even though the sample sizes in the analyses are relatively small ($n = 46$ for the forest experiment and $n = 92$ for the grassland experiment). In this situation AICc is better suited for model selection than AIC (Johnson & Omland 2004), as AIC tends to favour over-fitted models when sample sizes are small (Burnham et al. 2011). This also seems to be the case in the present study, as many marginally non-significant paths are retained in the final path models, that may have been excluded when applying more conservative selection criteria (Figs, 1 and 2).

-We agree with the Reviewer's comment, and accordingly used AICc (instead of AIC) in the re-analyses of our data. As expected, this led to the exclusion of some marginally significant pathways, but had no effect on the overall results and interpretation of the data.

Specific comments:

Discussion

Line 208: It is better to avoid statements like ‘;for the first time’.

-We agree and have removed “for the first time”.

Lines 272-276: Here the authors claim that of the many potential pathways, indirect effects via arthropod abundance were particularly important. It is unclear what the authors mean by important. Is it the strength of the indirect relationships or the fact that the effect of arthropod abundance on richness consistently appeared throughout the models? Figures 1 and 2 suggest that the second alternative more closely reflects the results of the SEMs.

-The Reviewer is right – the indirect effects are not always stronger than direct effects (in particular for herbivore species richness, which is then also reflected in the path models on overall arthropod species richness). Thanks to the suggestion given above for adding an additional figure (Figure 3), we can now show more clearly the relative importance of the direct and indirect(via abundance) effects of the different components of plant diversity on arthropod species richness. From this figure, it becomes clear that indirect effects make important contributions and that these effects can be as strong as or even stronger than the direct effects of plant diversity on arthropod richness (in particular for predators and parasitoids). To avoid misunderstandings, we have rephrased the sentences that summarize these findings (e.g. changing “largely explained” by indirect effects to “in many cases explained” by indirect effects; e.g. L48, 229). However, this does not change our conclusion that understanding effects of plant diversity on arthropod abundance can greatly help us to understand the mechanisms underlying plant diversity effects on arthropod species richness.

Lines 286-290: I very much like this example of how the results could inform conservation.

-Thank you. We agree that our findings can be relevant for conservation and hope that our results stimulate further research on these understudied issues.

Methods:

Lines 366: What were the criteria for classifying taxa as ‘;sufficiently sampled’ in the forest experiment?

-“Sufficiently sampled” was supposed to mean that for these taxa the employed methods are well accepted sampling strategies (e.g. beating for spiders and beetles, trap nests for Hymenoptera). We deleted this part of the sentence, as we mention this fact already at the beginning of the section on arthropod sampling (L392-396).

Lines 367 & 381: While the taxonomic groups considered in the two experiments do partly overlap, some of the groups were only considered in one of the experiments (e.g., Blattodea, Mantodea, Psocoptera and Lepidoptera were only considered in the forest experiment, whereas Isopoda, Myriapoda and Chilopoda were only considered in the grassland experiment). To what extent can the differences in the patterns between the two experiments/ecosystems be explained by differences in the composition of the analysed taxonomic groups?

-We agree that differences in the composition of arthropods can play a role in causing differences among the study systems, but we consider these compositional differences as a biological effect rather than a methodological effect. In each experiment, we focused on the dominant groups of organisms, which are different in grasslands and forests, but which should reflect the respective composition of herbivores, predators, and parasitoids in each system. We have added a note to the Methods to explain why the set of taxa did not completely overlap between the experiments (L436-440):

“We note that the set of organism groups considered in the two experiments is not identical. This is due to differences in arthropod communities of the two different types of ecosystems considered (forest vs. grassland), which nevertheless reflect the general composition of the dominant groups of herbivores, predators and parasitoids in each ecosystem.”

We now also mention this issue explicitly in the Discussion and refer to the Methods for further details (L249-251):

“with consequences also for arthropod community composition, as reflected by differences in the frequency and identity of dominant arthropod taxa; see Methods”

Lines 390-394: The sampling intensity (i.e., the number of trees sampled) differed between plots in the forest experiment. As mentioned earlier, it would be valuable if the authors could show by using alternative approaches (asymptotic richness estimators or rarefied species richness) that their results are robust. This is particularly important, because the abundance and species richness of consumers are the two primary response variables in the paper and the main claims are based on patterns related to these two variables.

-As suggested, we have added analyses based on rarefied arthropod species richness (cf. Supplementary Tables 18-24). These analyses underline the role of arthropod abundance in modifying the relationships between plant and arthropod diversity, as we explain above (please see our reply to the first major comment).

Line 430: This sentence either contains a repetition, or there might be a full stop missing after ‘;Leaf toughness and silica concentration were assessed.’.

-Yes, there was an unnoticed repetition that we have now deleted.

Lines 509-516: Here the authors treat the structural diversity variables of the two experiments differently, which raises the question to what extent differences in the importance of structural diversity between ecosystem are driven by differences in the treatment of the respective variables. Do the authors obtain similar results regarding the effects of the structural variables in the grassland experiment, if the two variables in this grassland experiment are also subject to PCA prior to the analyses?

-Subjecting the structural variables to a PCA in the grassland experiment is not sensible, because there are only two variables (so that PCA cannot be used for dimension reduction here and does not provide useable outcome). In the forest experiment, there were four variables, because structure was defined here as the combined effect of tree height and crown projection area. A variable comparable to crown projection area was not available for the grassland experiment and would probably not be as informative as in the forest experiment, where larger-scale structural variation is strongly determined by horizontal and vertical tree dimensions (the horizontal variation would be below the spatial resolution of our measurements of plant structure in the grassland experiment in most cases). We therefore consider differences in the way the structural variables are treated in the analyses as a reflection of differences in larger-scale spatial variability determined by the different growth forms of trees vs. herbs or grasses.

Line 536: Including residual covariance terms into path models only to improve model fit is not a well-justified argument. Residual covariance terms are included into structural equation models to account for common (but potentially unmeasured) sources of variance among the variables in a model. That is, covariance terms should be included if there is some a priori information about a potential correlation between two variables for which the causal structure/direction is unclear or unknown. Apart from that, I agree with the authors that the models should certainly contain residual covariance terms. I was wondering, however, why these terms were only included in the grassland models and why only certain covariance terms were considered. Were some covariance terms not considered because, the explanatory variables were extracted from PCAs? Did the authors exclude certain covariance terms based on AIC as for the directed paths? The criteria for including or excluding covariance terms should be described more clearly.

-We have now standardized the way in which covariances were included in the path models. We started with a full set of possible covariances between plant diversity variables (assuming that they are all not completely independent from each other) and removed non-informative covariances during the stepwise model simplification procedure. We have added a description to the Methods (L613-617):

“We additionally tested for significant residual covariances between the plant-based predictors (see Supplementary Fig. 3), as the different components of plant diversity might not be completely independent. We sequentially dropped non-informative pathways and covariances, if their removal resulted in a reduction of the second-order Akaike Information Criterion (AICc) of the models^{16,70}”

References:

Allan et al. (2015). Land use intensification alters ecosystem multifunctionality via loss of biodiversity and changes to functional composition. *Ecology Letters*, 18 (8), 834–843. doi:10.1111/ele.12469

Burnham, K. P.; Anderson, D. R.; Huyvaert, K. P. (2011). AIC model selection and multimodel inference in behavioral ecology. *Behavioral Ecology and Sociobiology*, 65, 23–35. doi:10.1007/s00265-010-1029-6

Castagneyrol, B., & Jactel, H. (2012). Unraveling plant–animal diversity relationships: a meta-regression analysis. *Ecology*, 93 (9), 2115–2124. doi:10.1890/11-1300.1

Hallmann et al. (2017). More than 75 percent decline over 27 years in total flying insect biomass in protected areas. *PLOS ONE*, 12 (10), e0185809. doi:10.1371/journal.pone.0185809

Hsieh, T. C., Ma, K. H., & Chao, A. (2016). iNEXT: an R package for rarefaction and extrapolation of species diversity (Hill numbers). *Methods in Ecology and Evolution*, 7 (12), 1451–1456. doi:10.1111/2041-210x.12613

Johnson, J. B., & Omland, K. S. (2004). Model selection in ecology and evolution. *Trends in Ecology & Evolution*, 19 (2), 101–108. doi:10.1016/j.tree.2003.10.013

Sandom et al. (2013). Mammal predator and prey species richness are strongly linked at macroscales. *Ecology*, 94 (5), 1112–1122. doi:10.1890/12-1342.1

Scherber et al. (2010). Bottom-up effects of plant diversity on multitrophic interactions in a biodiversity experiment. *Nature*, 468 (7323), 553–556. doi:10.1038/nature09492

Schuldt et al. (2015). Multitrophic diversity in a biodiverse forest is highly nonlinear across spatial scales. *Nature Communications*, 6 (1). doi:10.1038/ncomms10169

Storch, D., Bohdalkov, E., & Okie, J. (2018). The more-individuals hypothesis revisited: the role of community abundance in species richness regulation and the productivity–diversity relationship. *Ecology Letters*, 21 (6), 920–937. doi:10.1111/ele.12941

Reviewer #2 (Remarks to the Author):

The Authors tackle a highly interesting question, i.e. the role of structural and functional plant diversity in shaping the higher trophic levels (arthropod communities). To answer this, they measured specific plant traits and collected a high number of arthropod specimens, in experimental plots set in two different ecosystems, a subtropical forest in China and a grassland in Germany, they calculated functional and structural diversity indices, and they used path analysis in order to test the relationships between these indices and arthropod abundance in the plots of the examined ecosystems.

I love the concept of this study, more specifically:

1. The use of two different ecosystems is important for detecting/describing trade-offs in the functional roles of plant traits, according to local conditions.
2. I appreciate their extensive fieldwork samplings; I understand how painful this process can be, and I acknowledge their effort (especially given all the methodological implications included in it).
3. Sampling multiple trophic levels and functional guilds can offer a holistic view of the community.
4. Structural plant diversity as a predictor of arthropod diversity is indeed an overlooked factor.
5. Sampling multiple plant traits can also be painful, yet it allows a thorough exploration of interspecific relationships in study communities.

-Thank you for the constructive review and for the positive words! We fully agree.

However, I'm afraid that I do not agree with the Authors claim that, with their current analytical approach, they really disentangle the drivers of arthropod diversity regarding plants. For example, after reading the Discussion, I was left wondering if leaf trait diversity is really insignificant. But then again, why would a leaf trait diversity index be significant in describing one or two ecosystems? What would be the practical significance of this result, and how could it be used for restoration or conservation to tackle specific problems regarding diversity of the different trophic levels? On the other hand, the evaluation of specific traits could really guide us to conclusions useful for designing management schemes, or even help sketch the phenotypic profile of the functionally important species in the two ecosystems studied. These are topics that I missed from the Discussion, and I think that they cannot be answered with the current analytical scheme. Finding relationships between diversity indices, (a) perhaps is not novel enough, and (b) can hardly provide details on the mechanistic dimension of the relationships between plant and animals (it would probably do so on a macroecological scale, but here we have only two experimental study areas).

Again, as for the leaf traits, maybe there are indeed specific ones that matter for (i.e. predict the diversity of) the different trophic levels or functional guilds. Have you tried testing the mean values of leaf traits against arthropod diversity? Functional diversity indices are very useful, but I am afraid that in this particular case, alternative analyses are needed in order to actually disentangle the role of the components of plant functional diversity that predict arthropod diversity in the communities studied.

I really think that this is an amazing dataset, and most importantly, it allows for the application of many different analytical methods. There is, for example, the possibility of using Multivariate-response GLMs, and test each plant trait (or multiple predictors) against the entire community of the arthropods captured in each area.

To return explain this lack of significant relationships, the Authors argue that there might be more traits that have not been considered in this study, however, I doubt this.

Have the Authors tested this hypothesis?

-Thank you for the comments and the interesting suggestions for further analyses. First of all, we would like to point out that we tested for the effects of specific traits, as these were reflected in the PCA of the community-weighted mean values of five different traits (leaf toughness, LDMC, SLA, leaf N, leaf C, and in addition in the grassland experiment, silica concentration). However, we refrained from testing each trait individually because of covariation between traits and because we were particularly interested in the combined effects of these traits. Nevertheless, we interpreted effects of the PC axes of trait composition in terms of their loadings throughout the manuscript. Overall, there were only minor effects of

the PCs of community-weighted mean values on arthropod abundance and richness, such as on grassland herbivore, predator and parasitoid species richness (Fig. 2d-f), while they played almost no role in the forest experiment. Although we did not test the underlying traits directly, we can safely assume that they would not have more effect on arthropod abundance and richness than the PCs summarizing their combined effects. In addition to testing trait identity effects, we used functional diversity to test whether trait variation promotes arthropod species richness. Our main objective was to quantify the relative effects of different components of plant *diversity* on arthropod diversity, and our intention behind using functional diversity as an index based on multiple traits was to take into account the multidimensionality of functional trait space, where traits often do not act independently from each other and therefore differences in functional trait space among plant species may only become evident when multiple traits are considered simultaneously (see e.g. Kraft et al. 2015 <https://www.pnas.org/content/112/3/797> and Chauvet et al. 2017 <https://besjournals.onlinelibrary.wiley.com/doi/10.1111/1365-2435.12847>). Many papers have shown that multivariate functional diversity indices provide meaningful insights into how biodiversity affects ecosystems (see e.g. the thousands of citations to the Villéger et al. 2008 <https://esajournals.onlinelibrary.wiley.com/doi/10.1890/07-1206.1> and Laliberté & Legendre 2010 <https://esajournals.onlinelibrary.wiley.com/doi/10.1890/08-2244.1> key papers on multivariate FD). In order to show how expanding the framework of such previous studies can improve our understanding of biodiversity relationships, using comparable metrics as those of previous studies (species richness, functional diversity) will be most intuitive. We believe that it is very relevant information for designing management schemes to explore which biodiversity facets are crucial for supporting a high diversity of invertebrates.

We also agree that it would be rewarding to analyze the effects of trait identity and functional diversity on the arthropod community composition rather than on arthropod abundance and richness. However, this would be a completely different paper, with different methods and different approaches. Here, multivariate response GLMs might be a good way to explore these relationships. However, these interesting ideas are beyond the scope of our current study, which sets out to quantify the relative effects of different components of plant diversity on arthropod diversity – an approach not taken by previous studies, which usually focused on individual diversity components.

To further address the Reviewer's advice, we now include and discuss the relationships between plant species richness, functional diversity, and phylogenetic diversity in our manuscript (see also reply to comment below). The strong correlations between functional and phylogenetic diversity indicate that our functional diversity metrics capture overall trait space variation quite well (assuming that trait variation is phylogenetically conserved). This means that we in fact consider key functional traits, which has implications for the interpretation of our results. We have added this information to the Discussion (L277-280):

“While the lack of effects of leaf trait functional diversity on forest herbivores could indicate that functional traits not considered in our study play a role, strong correlations between functional diversity and phylogenetic diversity (often used as a metric to capture unmeasured variability in functional traits⁴⁴, see Methods) suggest that the traits used in our analyses account for an important part of the overall trait space.”

Moreover, what I felt was missing from this approach is the role of phylogenetic diversity of both the plant and the animal assemblage. Multispecies trait-based analyses benefit from assessing the role of phylogeny in predicting the distribution of traits among the members of the community, because type I errors due to phylogenetic similarity (phylogenetic pseudoreplication) are avoided.

The way that the questions of this analysis are currently structured, phylogenetically-corrected models (PGLS, PGLMM, etc.) are probably not applicable. However, is the phylogenetic dissimilarity of species within plots, a factor linked to arthropod species distribution? I understand that the vegetation plots are experimental (i.e. planted, not entirely natural); still, similarities in plant traits or metabolites (which are related to phylogeny and are not measured here) could potentially drive the composition of arthropod communities, as suggested by the Authors in L246-247.

Pairwise phylogenetic dissimilarity matrices are easy to generate both for plants (e.g. Phylomatic software) and for the animals --especially given that DNA barcoding has been used for verifying taxonomic identifications of the arthropod specimens.

-We agree with the Reviewer that phylogenetic diversity is an important component of plant diversity. Phylogenetic pseudoreplication should not be an issue for our analyses, because we look at aggregated plot-level data and not at comparisons among species. We have data on phylogenetic relationships available for the plant communities in both experiments, but unfortunately not for arthropods (although DNA-barcoding was used for some specimens in the forest experiment, this was not the case for most species and for the grassland experiment). We actually calculated phylogenetic diversity for the plant communities (based on Rao's Q , which is equivalent to MPD in the abundance/biomass-weighted case). However, phylogenetic diversity was highly correlated with functional diversity (and with plant species richness), indicating that there is not much additional information that can be gained by including phylogenetic diversity in the models. Most of its variation is already explained by functional diversity and plant species richness, and hypotheses based on the latter two are more developed and can be better addressed than hypotheses based on phylogenetic diversity as a proxy of functional trait space. Because models were already quite complex, we therefore decided not to consider phylogenetic diversity as an additional predictor. We have now added this information to the Methods (L521-534):

“Because analyzing selected traits might not necessarily capture the full variation in functional diversity, we additionally calculated plant phylogenetic diversity. Phylogenetic diversity might be used as a proxy of overall functional trait space if functional traits show a phylogenetic signal⁴⁴. We used ultrametric phylogenetic trees available for both experiments^{66,67} and calculated phylogenetic diversity, analogous to functional diversity, as biomass-weighted Rao's Q . However, functional and phylogenetic diversity were highly correlated in both experiments (Pearson's $r = 0.83$, $df = 44$, $P < 0.001$ for the forest experiment, and $r = 0.78$, $df = 90$, $P < 0.001$ for the grassland experiment, based on log-transformed values). The same applied to the relationship between phylogenetic diversity and plant species richness ($r = 0.82$, $P < 0.001$ in both experiments, df as above), whereas functional diversity was less strongly correlated with plant species richness ($r = 0.69$, $P < 0.001$ in both experiments, df as above). To avoid overly complex models, we therefore did not include phylogenetic diversity in our analyses, as its variation was already well reflected by functional diversity and plant species richness.”

To sum up, although I love the concept of this work, I have two main concerns: First, the use of functional diversity indices, instead of the traits measured, and second, the fact that phylogeny is omitted from the analyses.

-Thank you. As explained above, we more clearly present and discuss both, plant traits and phylogenetic diversity, in the revised version of the manuscript.

In other respects, this is a very well written and structured manuscript, with excellent use of the English language, presenting methodology in a clear and comprehensive way. My specific, line-by-line comments are minor:

-Thank you.

INTRODUCTION

- The Introduction is excellent, presenting a thorough review of the relevant literature, and clearly explaining the aims of the study.

L61-62: "Mixed support" is not a very clear term.

-Changed to "previous studies ... varied in their support ..." (L63).

L108-111: I am not sure that I entirely understand the Authors' distinction between the direct and the indirect effects of plant diversity to arthropod diversity at this point of the text. I would thus recommend rephrasing.

-We rephrased this section to better explain the meaning of direct and indirect effects (L110-117):

"Importantly, we specifically differentiated between direct and indirect plant diversity effects on arthropod species richness. We considered direct effects as those directly linking up plant diversity and arthropod species richness (e.g. because plant diversity-mediated habitat diversity provides more niches that support a higher diversity of arthropods³⁶). Because we hypothesized that arthropod species richness is influenced by changes in arthropod abundances (i.e. assuming that species richness is affected via 'more individuals'³⁷), we considered effects of plant diversity that modified arthropod abundances as indirect effects on arthropod species richness."

RESULTS

- Perhaps change some decimal separators in the text, from comma to dot?

-We have carefully checked the Results section to make sure that decimal separators are always dots. The commas used at the beginning of the section were intended to separate numbers > 1000. To avoid misunderstanding, we have now replaced these commas by a thin space.

- Although I understand the style of presenting results of path analysis models, I am afraid that the figures are way too complex. Perhaps I would remove the clipart, in order to make the figures less dense and crowded to the eye. Also, please mind that the current brown boxes cannot be distinguished from the pink ones by a person with color-blind vision.

-We now provide the figures without clipart and with changed colors. Moreover, we have added a third figure based on the recommendations of Reviewer 1, which helps to summarize the complexity of the relationships shown in the path models.

L767-768: Grammar

-Corrected.

L768: Can you explain "endogenous" variables?

-We have added "(dependent)" to explain that endogenous variables are dependent on other variables in the model.

Fig. 1: The explanation of the c) section of the figure is not clearly noted (it starts in L766?).

-Thanks for noticing. You are correct and we replaced "a) and b)" in the relevant part of the figure legend with "c" so that it can now be clearly identified.

METHODS

- Were there any pollinators collected with the suction method? Are there mutualists in the dataset --I read about parasitoids to Hymenoptera but are there parasitoids to herbivores?

-We did not analyze pollinators as a separate functional group (in addition to herbivores, predators, and parasitoids) because our sampling methods in general were not directly aimed at assessing pollinators and analyses would therefore be unreliable. We have added this information to the Methods (L421-423):

"Pollinators were not considered as a functional group in our analyses, because the sampling methods employed did not allow for a consistent assessment and reliable comparison between the experiments."

Parasitoids were all parasitic Hymenoptera sampled by suction sampling in the grassland experiment (i.e. including those of herbivores). We have added this to the Methods (L435). In the forest experiment, parasitoid assessments were based on trap nests, which provide data on a subset of parasitoids, as described in the text.

- I guess it is already included in the Supp. Material (although I couldn't have access to it), but I would like to see a list of the species sampled, and their classification into groups (predators, parasitoids, etc.).

-We have added this information as supplementary information (Supplementary Data 1).

L430: Please delete "Leaf toughness and silica concentration were assessed"

-Done.

L519-520: Please define direct and indirect.

-We added a short explanation (L595-587):

"We used path analysis⁷⁰ to assess the direct (paths from plant diversity to arthropod richness) and indirect (paths via arthropod abundance) effects of taxonomic, functional, and structural diversity of the plant communities on arthropod species richness."

This summarizes the more detailed explanation on direct and indirect effects that we have added to the Introduction (L110-117).

Reviewer #3 (Remarks to the Author):

Schultz et al. Multiple plant diversity components drive consumer communities across ecosystems.

Brief summary

This manuscript evaluates the relative importance of several mechanisms by which plant diversity affects arthropod communities in two different ecosystems, a temperate grassland and a subtropical forest. Specifically, the authors use path analyses to compare the effects of plant species richness, biomass, trait composition, functional diversity, and structural diversity (horizontal variability and vertical stratification) on total arthropod abundance and total arthropod species richness, as well as for three trophic levels separately (herbivores, predators, and parasitoids). They conclude that plant species richness has a consistent and positive effect on arthropod species richness, which is mediated by greater plant structural and functional diversity, however the specific mechanisms differed between ecosystems and trophic levels.

In general, the manuscript is well-written and would be relevant to a broad audience. The effects of plant diversity are of significant interest in ecology, however, this field has been limited to one or two trophic levels, and rarely have studies documented the separate trophic groups as done here. The described dataset is substantial in terms of the arthropod sampling and identification, breadth of traits measured, and structural components included in the analyses. However, the generalities that can be taken from the comparison of the grassland and forest ecosystems are limited due to the low sample size within each system, difference in climate, and general differences in experimental design (and perhaps more could be said to this end). Regardless, the combined dataset is impressive and highlights the complexities of biodiversity effects from a community-perspective.

-Thank you for taking the time to review our manuscript and for providing very helpful suggestions for improvement! We agree that more studies will be required to support the generality of our findings, and we now mention this explicitly in the Discussion (L268-271): *“While studies replicated across a wider range of environmental conditions and manipulative experiments will be required to verify the causal drivers and generality behind the observed effects, ...”*

Nevertheless, we are positive that our comparison of two large-scale experiments in very different types of ecosystems can provide new insights and hypotheses that benefit from the approach we have taken.

In measuring plant functional traits and diversity, the complete sampling design and resolution of the data is somewhat unclear. Were plant trait data taken for the same plots and individual trees that arthropods were sampled? Based on lines 422-428, trait values for trees are based on 5 individuals per species, while in the grassland traits were measured from 3 individuals per plot? Lines 422-423 and lines 428-429 are conflicting. If trait data are available at the individual plant or plot-level, it seems that these data should be used to relate

directly to arthropod data from the corresponding individuals/plots, rather than species-level means which may eliminate variance that is highly relevant to the sampled communities.

-We agree that it would be ideal to have trait data for each plant species on each study plot. Unfortunately, such data are only available for some of the traits measured in the grassland experiment. Other traits in the grassland experiment and all traits measured in the forest experiment were measured at a coarser scale (several individuals per species taken from different plots, but not from all plots). To keep the level of analysis comparable between the two experiments and among the different plant traits within each experiment (which was necessary to calculate functional diversity and trait composition), we aggregated trait data at the species level. This is the level used by most studies analyzing functional traits, and many studies even rely on data bases with third-party trait data from different geographic locations. So while our approach is necessarily limited in the degree of data resolution, an advantage is that the data were actually measured *in situ* on plants from the two experiments. We have checked for potentially conflicting information and added a sentence to the section on functional traits that clarifies the resolution of the data (L483-486):

“For both experiments, we used mean trait values per plant species as the average of trait measurements on individual plants collected in situ, because plot-level data were not available for any of the traits in the forest experiment and for several of the traits measured in the grassland experiment.”

Overall the statistical approach is solid, although the sample size (plots, see above) could be argued to be low for the complexity of the models being tested. In particular, the forest analysis is based upon a sample size of 42 (plots); is this sufficient to assess the inter-relationships among 9 variables? I’m not sure. In addition, the results are somewhat overwhelming due to the number of path diagrams (8) and separate tests for each trophic group as well as all arthropods together in both the forest and grassland. While I think the results from each of these individual tests are interesting, it is hard to digest with the short format of the article, where there is not enough space to discuss in depth the different hypotheses related to each test. Moreover, to consider each trophic level separately ignores any potential indirect effects of plant diversity mediated by co-occurring arthropods, like density- or diversity-dependent responses of natural enemies to herbivores. This is likely important, and what would be interesting is to compare the strength of such herbivore-mediated effects to the direct effects of plant diversity, although this of course adds complexity.

-We have added a new figure, as suggested by Reviewer 1, that summarizes the main results of the path models and makes them easier to grasp (Fig. 3). We hope that this new figure also addresses the issues raised above. Sample size in path models is debatable, and different authors make different recommendations. We agree that sample sizes are at the lower bound, but consider them sufficient for our rather exploratory approach. Adding another level of complexity by including all trophic levels in one model (e.g. to test for effects of herbivores on predators) would indeed be highly interesting. However, models would then definitely become too complex statistically (each trophic level would need direct pathways from plant diversity and indirect pathways from the lower trophic levels, multiplying the number of paths fitted in one model) and visually. Our models can only implicitly include these effects across higher trophic levels. For example, if herbivores modify plant diversity effects on predator diversity, this will be reflected in the strength of the pathways connecting plant diversity and predator diversity.

I am also concerned that In addition, the relationship arthropod species richness and abundance is directional but it could be argued that this should be treated as a covariance. It is also not clear which other covariances were included in the path models. The path between plant species richness and trait composition is omitted from the analyses, with the exception of plant species richness and trait composition PC2 in the grassland only. Is it accurate to say tree biomass causes structural diversity, or that these covary?

-See also reply to comments of Reviewer 1: We have added an alternative analysis where arthropod abundance and arthropod species richness are linked up via a covariance term. Comparison with our initial modeling approach showed that directional paths from abundance to arthropod species richness were generally more strongly supported (lower AICc) than a covariance structure. This lends strong support to our assumption that effects of arthropod abundance on arthropod species richness are highly relevant in explaining plant diversity effects on arthropod species richness. We also explain the theoretical expectations for these alternative models in the Introduction (L119-122):

“We also tested the alternative hypothesis of reciprocal interactions between arthropod species richness and abundance³⁸, which might be better reflected by residual covariance terms than by a directional pathway in the path models.”

We now also standardized the way in which covariances between predictors were included in the path models. We started with a full set of possible covariances between plant diversity variables (assuming that they are all not completely independent from each other) and removed non-informative covariances during the stepwise model simplification procedure.

We have added a description to the Methods (L613-617):

“We additionally tested for significant residual covariances between the plant-based predictors (see Supplementary Fig. 3), as the different components of plant diversity might not be completely independent. We sequentially dropped non-informative pathways and covariances, if their removal resulted in a reduction of the second-order Akaike Information Criterion (AICc) of the models^{16,70”}

Finally, there is some context for the novelty of the results presented here that isn't addressed. For example, there is a large literature on the Enemies Hypothesis showing the linkage between plant diversity and predator abundance and diversity. And its well understood that structural complexity affects predator abundance / diversity by reducing intra-guild predation. Some of the relevant literature on these topics is cited, but the text doesn't really make clear that some of the results presented here are anticipated or previously demonstrated by this literature.

-We agree with the reviewer and carefully checked the text to see where more explicit reference to previous work might be required. We added a sentence in the Discussion with direct reference to the 'enemies hypothesis' and its relevance for our findings (L284-287):

“This is in line with the assumption of the 'enemies hypothesis' that effects of plant diversity on predator diversity can also operate via modifications of habitat structure or reduction of intra-guild predation^{14,23”}

If we have missed other important sections where reference to specific previous work seems reasonable, we would be very grateful for further recommendations. While several papers have dealt with effects of plant structure on higher trophic levels (the most relevant of which we cite in the manuscript), there is a lack of studies that have actually analyzed the relevance of these effects in the context of changing plant diversity. We mention this earlier in the Discussion and cite relevant literature (L240-244):

“However, structural diversity as an additional moderator of plant diversity effects on consumer diversity has received much less attention in this respect. This is despite the well-known fact that plant structure significantly affects herbivores, predators, and parasitoids by modifying environmental conditions and habitat space²³⁻²⁵, and that plant species richness can influence the physical structure of plant communities^{41,42}.”

Lines 108-113: Do increases in abundance cause increases in richness? Clearly they are tied to one another

-We now make clear that it is our assumption (based on the “more-individuals” hypothesis) that arthropod abundance affects arthropod species richness, but that there are alternative ways to interpret the relationship between the two (covariation due to more complex interactions), which we now also address in our analyses (as suggested by the Reviewer). We have adapted this part of the Introduction accordingly (L110-122):

“Importantly, we explicitly differentiated between direct and indirect plant diversity effects on arthropod species richness. We considered direct effects as those directly linking plant diversity to arthropod species richness (e.g. because plant diversity-mediated habitat diversity provides more niches that support a higher diversity of arthropods³⁶). Because we hypothesized that arthropod species richness is influenced by changes in arthropod abundances (i.e. assuming that species richness is affected via ‘more individuals’³⁷), we considered effects of plant diversity that modified arthropod abundances as indirect effects on arthropod species richness. Our study therefore provides important insights into the potential mechanisms linking changes in plant communities to consumer diversity via changes in abundances. We also tested the alternative hypothesis of reciprocal interactions between arthropod species richness and abundance³⁸, which might be better reflected by residual covariance terms than by a directional pathway in the path models.”

Supplement

Table 2 – Shouldn’t PC1 inherently explain a greater proportion of variance than PC2?

-Yes. The last row in the table shows cumulative percentages of explained variance, i.e. summed effects for the two principal components in the case of PC2. For clarity, we have now added a row that shows the unique contribution of each PC.

Specific comments:

Line 85-87: Repetitive to previous sentence

-We replaced “how plant functional and structural diversity influence ...” by “plant diversity effects” to put a stronger focus on arthropod abundances, which are introduced in this sentence (L87-89) and further explored in the following sentences.

Line 90-92: Suggest dropping "consequences of such changes" to "consequences" and making a distinction between trophic groups. I am not sure this sentence describes an indirect effect of plant diversity, it is somewhat unclear. Is it meant that variation in plant diversity affects arthropod diversity through arthropod abundances? In the case of herbivores, would this not

be a direct effect of plant diversity, whereas for a predator or parasitoid it could be indirect if they are responding to herbivores (or direct if they are responding to plant structural complexity)?

-We dropped “of such changes” as suggested, and rephrased the sentence to better explain the effects we are talking about. It is correct that we define indirect effects of plant diversity on arthropod diversity as those effects that operate by modifying arthropod abundances (no matter whether this effect operates via multiple trophic levels, such as from herbivores to predators) or within a particular trophic level, such as herbivores). This is opposed to effects that directly link plant and arthropod diversity, e.g. because increased plant diversity provides more niches (in terms of structures and resources) for more arthropod species. We hope that the changes we have made make this more intuitively understandable (L92-95):

“However, the linkages between changes in plant diversity, changes in arthropod abundances, and the consequences for arthropod diversity (i.e. indirect effects of plant diversity that modify arthropod diversity via changes in arthropod abundances) at the scale of local communities are not yet well understood”

Line 98-100: Is a result general if detected in 2 vs 1 experiment? What about within-system variation? As stated in the introduction, there are mixed results from biodiversity experiments regarding plant diversity effects on higher trophic levels so I find it difficult to make this comparison between grassland and forest.

-We agree that our study can only provide a starting point for the discussion on the generality of plant diversity effects across ecosystems, and that more research is required that will hopefully be stimulated by our analyses. Nevertheless, we think our study approach (based on two comprehensive datasets acquired from two large-scale experiments in contrasting ecosystems) can provide important first insights that have not been addressed previously. We rephrased the sentence for clarity (L100-102):

“This comparison can help us to obtain first insights into the extent to which effects of plant diversity might operate in similar ways in contrasting ecosystems.”

Line 102-106: Too much in parentheses. The explanations of the calculations could come later. Hypothesis "related to microclimate and habitat space" introduced earlier and could be cut here. The specific components of functional diversity could be fleshed out more.

-We have removed the parentheses and now provide the information (with more details on the components of functional diversity) as additional sentences (L102-110):

“We use path models to analyze the relative contribution of direct and indirect effects of plant taxonomic diversity (species richness), functional diversity and composition, and structural diversity on overall arthropod species richness and the species richness of major trophic groups of arthropods (Supplementary Fig. 1). We quantified functional diversity as the variability among plant species in morphological and chemical leaf traits that were shown previously to affect arthropods^{22,33,34}. Because plant traits can further influence arthropods via mass-ratio effects³⁵, we also tested for the effects of mean trait values on arthropod abundance and species richness. Vertical stratification and horizontal variation of plant height within study plots were used to quantify plant structural diversity.”

Line 122-135: combine paragraphs?

-Done.

Line 133-135: Wording unclear - "correlations between abundance and/or richness values across higher trophic levels". Are these correlations between arthropod abundance and arthropod richness? Or between plants and arthropods?

-We rephrased the sentence to clarify that we mean correlations between arthropod data (L139-142):

“In both experiments, Pearson’s correlations between abundance and/or richness values of herbivores, predators, and parasitoids were always positive when significant ($P \leq 0.05$; Supplementary Table 1).”

Line 139: "either directly or indirectly via effects’..."

-Changed as suggested.

Line 144 & 146: "for the log-log relationship" unclear

-Changed to *“for log-transformed species richness data”* (L154/156).

Line 146: "tended" to "showed a marginal relationship" or similar

-Changed to *“Likewise, predator and parasitoid species richness showed a marginally positive relationship with plant species richness”* (L156-158).

Line 162-164: 171-175: Not clear what aspect of arthropods has an effect on, abundance?

-We added species richness and/or abundance where applicable.

Line 202: "between" not "among"

-Replaced accordingly.

Line 231-234: States there are differences in sampling methods but then says they are the same.

- Our intention is to say that although the exact sampling methods differ (beating vs. suction sampling, based on the best way to sample arthropods from trees vs. grassland plants), these methods nevertheless collected most arthropods directly from the vegetation. We have revised the sentence to avoid misunderstanding (L260-262):

“Differences in sampling methods between the study systems probably play a minor role: forest arthropods were all sampled directly from the vegetation (by beating), as were most grassland herbivores and parasitoids (primarily captured by suction sampling).”

Line 234-236: What is the argument for restricting the analysis?

-Ground-active arthropods might be less affected by plant structure than arthropods directly associated with the vegetation. Therefore, we tested whether results would change when only arthropods sampled directly from the vegetation were considered in the grassland experiment. We have added a short explanation for clarification (L263-266):

“Moreover, although most grassland predators were ground-active (sampled with pitfall traps), restricting the analyses to predators sampled from the vegetation (for which plant structure might be more important than for ground-active arthropods) did not change the relative importance of plant functional vs. structural diversity effects.”

Line 249-251: ‘This implies that leaf trait effects...not tree diversity’

-Replaced “rather than” by “not” (L283-284).

Line 310-314: If 46 plots were used, why mention 64? Why is there lower replication at higher levels of diversity?

Line 315-319: Less diverse = monocultures? The explanation of how plots were selected is unclear. What is meant by non-overlapping fractions? Why are the 24-species plots "additional"? Do the 24-species plots contain species not present in any of the other plots?

-We have rephrased this section to clarify why we mention the 64 plots and how the species composition of the mixtures was determined (L356-370):

" Our analyses followed the design for a set of 64 (32 per site, randomly distributed across the sites) ‘very intensively studied’ plots. Tree species composition of the mixtures was determined by randomly assigning (without replacement) each species of the 16-species mixtures to one 8-species mixture, subdividing these sets of eight tree species to non-overlapping subsets of four species, and the 4-species subsets to non-overlapping 2-species mixtures³². The 24-species mixtures were included as an additional high diversity treatment, which contained an additional eight species not present in the other plots of the study site. Tree species composition differed between the two sites, with two separate species pools of 16 broadleaved species in each site and an additional eight species shared between sites in the 24-species mixtures. All plots were weeded twice a year, with all upcoming vegetation between the planted trees being removed. For further details on the experimental design, see ref.³². Lack of or limited tree establishment (8 plots) and lack of arthropod sampling (10 plots, see below) limited the final set of plots to 46 (16 monocultures, 14 2-species mixtures, eight 4-species mixtures, four 8-species mixtures, two 16-species mixtures, and two 24-species mixtures). ”

Line 403: what does 6x6 and 12x12 mean? Do plots differ in the number of trees?

-No, the number of trees planted in each plot was the same. We agree that this was unclear. The numbers provided here show the number of trees used for the quantification of tree height and volume (i.e. 36 and 144, respectively). The numbers of trees analyzed per plot were higher for plots with four and more tree species to ensure that multiple individuals of each tree species were included in the assessments (which can be achieved by sampling less trees

in monocultures and 2-species-mixtures). Data were then standardized to reflect total plot biomass, so that values are comparable among all plots. Trees were planted in a grid of 20 x 20 individuals in each plot, and 6 x 6 and 12 x 12 refers to the grid positions in the center of each plot. We have added an explanation to the text (L463-465):

“Assessments were based on the central 6 x 6 trees per plot (out of the grid of 20 x 20 trees planted in each plot) in monocultures and 2-species mixtures, and the central 12 x 12 trees in more diverse mixtures.”

Line 430-431: "Leaf toughness and silica concentration were assessed Leaf toughness'...."

-We deleted the repetition.

Line 431-434: Mention of mesocosm plants is a surprise

-We now provide an explanation of why data from mesocosms (with plants taken from the experiment) were used for leaf toughness in the grassland experiment. Readers interested in more information on the design of the mesocosms are referred to the cited reference (L493-496):

“Data on leaf toughness was not directly available from plants grown in the field, but measured for five healthy and fully developed leaves on each of five replicate individual mesocosm plants (see ref.⁶²), grown in PVC pipes (15 cm diameter, 60 cm length) filled with sieved field soil from the Jena Experiment mixed with 20% sand.”

Line 455-458: Include this information earlier in main text as well

-We provide this information in the Results section, when the main effects of trait composition on arthropod abundance and species richness are described (L207-215).

Reviewers' Comments:

Reviewer #1:

Remarks to the Author:

I have read the revised version of the manuscript "*Multiple plant diversity components drive consumer communities across ecosystems*" by Schuldts and colleagues along with the authors' responses to my comments and the comments of the other two reviewers.

I am satisfied with the authors' responses to the comments and with the revision of the manuscript and I think the authors did an excellent revision job by providing additional analysis that support their original conclusions. I only have some minor comments, which I hope help to further improve the manuscript.

A) I support the authors' justification of the removal of singletons prior to the analysis in order to exclude vagrant species that might not be associated with the study systems (plots) and the strong correlations between arthropod species richness before and after the removal of singletons (> 0.97), increased my confidence in the approach. I also appreciate the additional rarefaction analysis the authors have used to substantiate their conclusion that changes in arthropod richness are mediated by arthropod abundance.

Yet, I recommend the authors to include their justification for the removal of singletons also in the respective part of the methods section (somewhere around line 444), so that readers from outside the field can follow the rationale behind this approach (i.e., a statement similar to the one in the response letter "*We removed singletons to make the data set more robust, because species recorded with just one individual in the whole data set might be vagrants that are not really associated with the study systems.*"). It would also be desirable if the authors could clearly state the number of individuals and species in each group (herbivores, predators and parasitoids) and system (grassland and forest) before and after removal of singletons. This will help readers assess the prevalence of singletons in the original data.

B) The authors have done a great job summarizing the results of their path analysis in Figure 3. Yet, there is one inconsistency between figure 3 and its legend. In the legend the authors state "*Effects are either direct (lighter hues, left bar of each diversity component, [...]) or indirect via arthropod abundance (darker hues, right bar of each diversity component, [...]).*" I guess the text should rather read "*Effects are either direct (darker hues, left bar of each diversity component, [...]) or indirect via arthropod abundance (lighter hues, right bar of each diversity component, [...]).*", because the left bars are coloured with the darker hues. Moreover, I guess the intention of the colour-code was to reflect the colour-code used for the different diversity components in the path models, which is a good idea. I had, however, problems to match the colours visually as they are not exactly the same. It would be great if the authors could extract the exact rgb-values of the colours used in the path models and then use these for the darker hues in Figure 3. The lighter hues could then be created based on these same colours.

C) In Supplementary Tables 2 to 24 the authors should check that the subheadings of each column are properly aligned with the content below (that is, if model estimates are right-justified, then the subheading of the respective should be as well).

Best wishes,

Jörg Albrecht

Reviewer #2:

Remarks to the Author:

The Authors have addressed all of the comments, and they have greatly improved the manuscript. A final suggestion, for clarity, would be to provide plant phylogeny in the Supplementary Material, as well as the values of the diversity metrics (functional, structural, phylogenetic) for the study plots.

Reviewer #3:

Remarks to the Author:

The authors have done an excellent job responding to both my own previous critiques (Reviewer 3) and it appears to me to the thoughtful comments of the other reviewers. There is obviously quite a bit that can be done with such a complex dataset, and there were a lot of suggestions for different approaches and further analyses. Reading all of the reviewer comments and the authors' responses, it appears to me they revised appropriately, and provided good justifications where the proposed changes were not made.

I have one substantive comment remaining. In my earlier review I commented on the lack of attention (in citations, analyses, etc.) to feedbacks among trophic levels. For example, plant on effects on herbivores will without question cascade up to indirectly affect predators, parasitoids, etc. Similarly, direct effects of plants on higher trophic levels will certainly cascade down to affect herbivores. I respect the authors statement that more complicated models may not be feasible with the given dataset, but the wording in the results and discussion is too definite on mechanism. What can be stated unambiguously are the demonstrated associations between plant and arthropod metrics, and then there is evidence for how effects on arthropod diversity can be mechanistically separated between indirect effects via abundance and direct effects. Acknowledging this can make the writing / wording challenging, the results language should be pushed towards more circumspect wording (like "association") rather than overly certain claims of mechanism. The Discussion gives more room for speculation. But currently I can see no mention of how trophic interactions among herbivores and predators. A vast literature tells us that herbivore abundance and diversity affects enemy (predator and parasitoid) communities and likewise that enemy communities affect herbivore communities. Furthermore, there are almost certainly also effects mediated by unstudied trophic levels, most notably insectivorous birds. But as far as I can tell there is no mention of these dynamics in the discussion or elsewhere in the manuscript. Its fine that the current dataset does not allow for disentangling these complexities, but the manuscript needs to acknowledge these important dynamics, and that some unknowable portion of the observed associations are due to these dynamics.

REVIEWERS' COMMENTS:

Reviewer #1 (Remarks to the Author):

I have read the revised version of the manuscript “*Multiple plant diversity components drive consumer communities across ecosystems*” by Schuldt and colleagues along with the authors’ responses to my comments and the comments of the other two reviewers.

I am satisfied with the authors’ responses to the comments and with the revision of the manuscript and I think the authors did an excellent revision job by providing additional analysis that support their original conclusions. I only have some minor comments, which I hope help to further improve the manuscript.

-Thank you once again for the very helpful comments that allowed us to clarify our approach and to improve the presentation of our main results!

A) I support the authors’ justification of the removal of singletons prior to the analysis in order to exclude vagrant species that might not be associated with the study systems (plots) and the strong correlations between arthropod species richness before and after the removal of singletons (> 0.97), increased my confidence in the approach. I also appreciate the additional rarefaction analysis the authors have used to substantiate their conclusion that changes in arthropod richness are mediated by arthropod abundance.

Yet, I recommend the authors to include their justification for the removal of singletons also in the respective part of the methods section (somewhere around line 444), so that readers from outside the field can follow the rational behind this approach (i.e., a statement similar to the one in the response letter “*We removed singletons to make the data set more robust, because species recorded with just one individual in the whole data set might be vagrants that are not really associated with the study systems.*”). It would also be desirable if the authors could clearly state the number of individuals and species in each group (herbivores, predators and parasitoids) and system (grassland and forest) before and after removal of singletons. This will help readers assess the prevalence of singletons in the original data.

-We agree that further clarification of this issue can be helpful, and we added the information requested by the Reviewer. In the relevant part of the Methods, we now explain why singletons were removed, and we added data on the percentage of singleton species at each trophic level for both systems to help readers to evaluate the prevalence of singletons in the data sets (L462-471):

“We removed singletons to make the data set more robust, because species recorded with just one individual in the whole data set might be vagrants that are not really associated with the respective study systems or the specific plots they were recorded in. While singleton species accounted for 13-49% of the total number of species across all study plots (forest: 47%, 48%, 49%, and 20% of all, herbivorous, predatory, and parasitoid species, respectively; grassland: 31%, 30%, 28%, and 13% of all, herbivorous, predatory, and parasitoid species, respectively), singleton removal did not influence overall patterns among study plots of arthropod species richness and abundance, which were highly correlated in the data sets with and without singletons (Pearson correlation, $r > 0.97$, $P < 0.001$ in all cases and for all trophic levels).”

B) The authors have done a great job summarizing the results of their path analysis in Figure 3. Yet, there is one inconsistency between figure 3 and its legend. In the legend the authors

state “Effects are either direct (lighter hues, left bar of each diversity component, [...]) or indirect via arthropod abundance (darker hues, right bar of each diversity component, [...]).” I guess the text should rather read “Effects are either direct (darker hues, left bar of each diversity component, [...]) or indirect via arthropod abundance (lighter hues, right bar of each diversity component, [...]).”, because the left bars are coloured with the darker hues. Moreover, I guess the intention of the colour-code was to reflect the colour-code used for the different diversity components in the path models, which is a good idea. I had, however, problems to match the colours visually as they are not exactly the same. It would be great if the authors could extract the exact rgb-values of the colours used in the path models and then use these for the darker hues in Figure 3. The lighter hues could then be created based on these same colours.

-We thank the Reviewer for carefully checking the new figure and for detecting this inconsistency! Indeed, the Reviewer is right and the darker hues are the ones for the left bars in each category, not the right bars. We have changed this as suggested by the Reviewer. We have also updated the color scheme, which in fact was supposed to reflect the colors used in Figs 1 & 2. As suggested, we now extracted the exact RGB values in Figs 1 & 2 and used these for the bars with darker hue in the new Fig. 3 (and bars with lighter hue based on the same colors).

C) In Supplementary Tables 2 to 24 the authors should check that the subheadings of each column are properly aligned with the content below (that is, if model estimates are right-justified, then the subheading of the respective should be as well).

-Again, a very helpful comment! We re-checked all tables and have taken care to align content and headings appropriately.

Best wishes,

Jörg Albrecht

Reviewer #2 (Remarks to the Author):

The Authors have addressed all of the comments, and they have greatly improved the manuscript. A final suggestion, for clarity, would be to provide plant phylogeny in the Supplementary Material, as well as the values of the diversity metrics (functional, structural, phylogenetic) for the study plots.

-We acknowledge the helpful suggestions by the Reviewer! We have added a new Figure to the Supplementary Information (Supplementary Fig. 2) which shows plots of the ultrametric phylogenetic trees of the plant communities of the two systems. Data on plot-level diversity metrics (functional, structural, phylogenetic) are included in the supporting data set uploaded at <https://idata.idiv.de/> (see Data Availability statement in the manuscript).

Reviewer #3 (Remarks to the Author):

The authors have done an excellent job responding to both my own previous critiques (Reviewer 3) and it appears to me to the thoughtful comments of the other reviewers. There is obviously quite a bit that can be done with such a complex dataset, and there were a lot of suggestions for different approaches and further analyses. Reading all of the reviewer comments and the authors' responses, it appears to me they revised appropriately, and provided good justifications where the proposed changes were not made.

- We thank the Reviewer again for their time and effort invested in reviewing and helping to improve the manuscript!

I have one substantive comment remaining. In my earlier review I commented on the lack of attention (in citations, analyses, etc.) to feedbacks among trophic levels. For example, plant on effects on herbivores will without question cascade up to indirectly affect predators, parasitoids, etc. Similarly, direct effects of plants on higher trophic levels will certainly cascade down to affect herbivores. I respect the authors statement that more complicated models may not be feasible with the given dataset, but the wording in the results and

discussion is too definite on mechanism. What can be stated unambiguously are the demonstrated associations between plant and arthropod metrics, and then there is evidence for how effects on arthropod diversity can be mechanistically separated between indirect effects via abundance and direct effects. Acknowledging this can make the writing / wording challenging, the results language should be pushed towards more circumspect wording (like "association") rather than overly certain claims of mechanism. The Discussion gives more room for speculation. But currently I can see no mention of how trophic interactions among herbivores and predators. A vast literature tells us that herbivore abundance and diversity affects enemy (predator and parasitoid) communities and likewise that enemy communities affect herbivore communities. Furthermore, there are almost certainly also effects mediated by unstudied trophic levels, most notably insectivorous birds. But as far as I can tell there is no mention of these dynamics in the discussion or elsewhere in the manuscript. Its fine that the current dataset does not allow for disentangling these complexities, but the manuscript needs to acknowledge these important dynamics, and that some unknowable portion of the observed associations are due to these dynamics.

-We agree with the Reviewer that more careful wording can help to avoid misunderstandings, and we have revised the Results and Discussion accordingly – replacing “effect” and wording that might be taken as overly certain claims of definitive mechanisms by more circumspect wording describing “relationships”, “associations” and by making clear that causalities and mechanisms are our interpretation of the statistical path models.

With respect to the discussion of potential interactions and feedbacks among higher trophic levels, we admit that these issues are not addressed in as much detail as might be preferred by some readers (but note that our Discussion touches upon these issues in several places, e.g. L264-268 and L291-295). This is largely due to space restrictions and the fact that our analyses cannot address all of these issues in detail (as explained previously and acknowledged by the Reviewer). However, and as suggested by the Reviewer, we have now added a statement in the concluding part of the Discussion that directly acknowledges these additional interactions and dynamics, how these issues relate to our findings, and how future research can further our understanding of the relationships between biodiversity and the trophic complexity of ecosystems. We also cite several papers as examples for key interactions between trophic levels, but did not provide a more extensive literature overview because this would have increased the number of references for the manuscript well beyond the limit of 70 recommended in the author guidelines (L340-346):

“In our study, these effects would have stayed elusive without the inclusion of plant structural diversity, highlighting the benefits of simultaneously considering multiple components of plant diversity and the potential mechanisms discussed above. The same may be true for higher trophic level diversity and the diversity of interactions among trophic levels, and we hope that our study stimulates future research exploring such interactions. Top-down effects of predators and parasitoids on herbivores¹⁴, cascading effects of plant diversity via herbivores on secondary consumers¹⁶, or effects of other functional groups (e.g. insectivorous birds⁵⁰) are additional modifiers that deserve further research and that our models take into account only implicitly by analyzing the net effect of plant diversity on individual trophic levels.”